# Widespread exposure to SARS-CoV-2 in wildlife communities

Amanda R. Goldberg [1], Kate E. Langwig [1], Katherine L. Brown [2,3,4], Jeffrey M. Marano [5,6], Pallavi Rai [5], Kelsie M. King [7], Amanda K. Sharp [7], Alessandro Ceci [4], Christopher D. Kailing [1], Macy J. Kailing [1], Russell Briggs [4], Matthew G. Urbano [4], Clinton Roby [4], Anne M. Brown [7,8,9,10,11], James Weger-Lucarelli [3,5], Carla V. Finkielstein [1,2,3,4,10,11,12] ✉ & Joseph R. Hoyt [1,12] ✉

Pervasive SARS-CoV-2 infections in humans have led to multiple transmission events to animals. While SARS-CoV-2 has a potential broad wildlife host range, most documented infections have been in captive animals and a single wildlife species, the white-tailed deer. The full extent of SARS-CoV-2 exposure among wildlife communities and the factors that influence wildlife transmission risk remain unknown. We sampled 23 species of wildlife for SARS-CoV-2 and examined the effects of urbanization and human use on seropositivity. Here, we document positive detections of SARS-CoV-2 RNA in six species, including the deer mouse, Virginia opossum, raccoon, groundhog, Eastern cottontail, and Eastern red bat between May 2022–September 2023 across Virginia and Washington, D.C., USA. In addition, we found that sites with high human activity had three times higher seroprevalence than low human-use areas. We obtained SARS-CoV-2 genomic sequences from nine individuals of six species which were assigned to seven Pango lineages of the Omicron variant. The close match to variants circulating in humans at the time suggests at least seven recent human-to-animal transmission events. Our data support that exposure to SARS-CoV-2 has been widespread in wildlife communities and suggests that areas with high human activity may serve as points of contact for cross-species transmission.

Severe acute respiratory syndrome coronavirus 2 (SARS-CoV-2), the causative agent of coronavirus disease 2019 (COVID-19), has resulted in over 771 million human cases and over six million deaths worldwide[1]. As SARS-CoV-2 becomes endemic in humans, one of the greatest threats to public health is the resurgence of more virulent and transmissible variants. The considerable pathogen pressure imposed by the pandemic has caused concern as to whether SARS-CoV-2 will spill into wildlife populations,

[1]Department of Biological Sciences, Virginia Tech, Blacksburg, VA, USA. [2]Virginia Tech Carilion School of Medicine, Virginia Tech, Roanoke, VA, USA. [3]Center for Emerging, Zoonotic, and Arthropod-borne Pathogens, Virginia Tech, Blacksburg, VA, USA. [4]Molecular Diagnostics Laboratory, Fralin Biomedical Research Institute, Virginia Tech, Roanoke, VA, USA. [5]Department of Biomedical Sciences and Pathobiology, Virginia Tech, Blacksburg, VA, USA. [6]Translational Biology, Medicine, and Health Graduate Program, Virginia Tech, Roanoke, VA, USA. [7]Program in Genetics, Bioinformatics, and Computational Biology, Virginia Tech, Blacksburg, VA, USA. [8]Department of Biochemistry, Virginia Tech, Blacksburg, VA, USA. [9]Data Services, University Libraries, Virginia Tech, Blacksburg, VA, USA. [10]Virginia Tech Center for Drug Discovery, Virginia Tech, Blacksburg, VA, USA. [11]Academy of Integrated Science, Virginia Tech, Blacksburg, VA, USA. [12]These authors contributed equally: Carla V. Finkielstein, Joseph R. Hoyt. ✉e-mail: finkielc@vt.edu; hoytjosephr@gmail.com

establish a sylvatic cycle, and potentially serve as a source for new variants.

Transmission of SARS-CoV-2 to captive animals has been well documented[2–4], but detections in free-ranging wildlife are currently limited to only a few species including white-tailed deer (*Odocoileus virginianus*[5–7]), feral mink (*Neovison vison*[8]), and Eurasian river otters (*Lutra lutra*[9]). Experimental infections and modeling of the functional receptor for SARS-CoV-2 (angiotensin-converting enzyme 2: ACE2) have shown that numerous wildlife species may be competent hosts[10–15]. However, it remains unexplored whether a diversity of wildlife species are infected in natural settings, where exposure to SARS-CoV-2 is likely to be indirect and at a lower exposure dose.

Since the emergence of SARS-CoV-2 in 2019, numerous variants have been detected in humans and animals. Many variants that have become dominant have mutations that increase their infectivity in humans[16], and may also impact the virus's ability to infect new wildlife species. SARS-CoV-2 collected from white-tailed deer have included lineages circulating in humans, caused by human-to-deer transmission[5], but have also included lineages with unique mutations suggestive of deer-to-deer transmission[17]. This implies that only minimal adaptation may be needed for transmission to occur among deer following initial human-to-animal transmission events[18]. Other human peridomestic species, such as deer mice (*Peromyscus maniculatus*[12,13]) and skunks (*Mephitis mephitis*)[14] have been shown to be capable of viral shedding in laboratory settings[11]. Collectively, these studies raise important questions about the extent of human-to-wildlife transmission and the ability of other wildlife species to sustain transmission.

Establishment of SARS-CoV-2 infections in wildlife communities could result in novel mutations that increase virulence, transmissibility, or confer immune escape, negatively impacting both human and wildlife populations. Furthermore, as SARS-CoV-2 adapts to not only human hosts, but potentially a wide diversity of wildlife species, SARS-CoV-2 evolution may become more unpredictable[19]. This could present several challenges for human health, including concerns related to vaccine development targeting human-specific lineages, and novel impacts to pathogenicity and transmissibility of the virus.

Here, we examine how widespread SARS-CoV-2 exposure has been in wildlife communities between May 2022 and September 2023. We used quantitative reverse transcription polymerase chain reaction (RT-qPCR) to examine 789 nasopharyngeal/oropharyngeal samples from 23 species sampled across Virginia and Washington D.C., USA and documented the presence of SARS-CoV-2 RNA in six of these species. In addition, we analyzed 126 serum samples from six species collected before and after the arrival of SARS-CoV-2 and detected neutralizing antibody titers in five of the six species. Finally, we detected an effect of urbanization and human use on seropositivity in animals, and examined genomic data associated with positive samples.

## Results
### SARS-CoV-2 RNA detections in multiple species
We amplified SARS-CoV-2 RNA extracted from nasopharyngeal/oropharyngeal samples by RT-qPCR from six of the 23 species examined (26.1% of species sampled) and had a total of 23 unique individual animals that were positive (2.9% of samples tested; *n* = 789; Fig. 1a, b, Supplementary Tables 1–3, Supplementary Data 1). This included eight deer mice (*P. maniculatus*; 4.7%, *N* = 172), four Virginia opossums (*Didelphis virginiana*; 2.9%, *N* = 140), four raccoons (*Procyon lotor*; 4.8%, *N* = 84), three Eastern cottontail rabbits (*Sylvilagus floridanus*; 2.5%, *N* = 118), three groundhogs (*Marmota monax*; 9.7%, *N* = 31), and one Eastern red bat (*Lasiurus borealis*; 8.3%, *N* = 12) (Fig. 1b). We had slightly higher positivity rates in field collected samples compared to samples collected at wildlife rehabilitation centers (4.04% compared to 2.24%), which may reflect repeated sampling of wildlife at a site during a SARS-CoV-2 outbreak.

### SARS-CoV-2 neutralizing antibodies in mammal communities
We detected SARS-CoV-2 antibodies in five of the six species we collected serum samples from in 2022 (60% neutralization cutoff, Fig. 1c, d and Supplementary Data 2), including the Virginia opossum (37.5%, *N* = 8), raccoon (36.4%, *N* = 11), Eastern gray squirrel (*Sciurus carolinensis*; 57.1%, *N* = 7), white-footed mouse (*Peromyscus leucopus*; 16.7%, *N* = 6) and the deer mouse (7.1%, *N* = 14). Percent neutralization values from samples prior to the arrival of SARS-CoV-2 (pre-2020) were significantly lower than those collected after SARS-CoV-2 arrival (*t* = −10.774, *p* < 0.001, Fig. 1d). Furthermore, four samples (two raccoons, one opossum, and one white-footed mouse) had a percent neutralization above a more conservative 80% cut-off, further supporting that previous SARS-CoV-2 exposure is likely in these species (Fig. 1d). In 2022, an opossum with a positive detection was trapped one month later and collected serum revealed a percent neutralization value of 51.1% which was below our 60% cutoff, suggesting this cutoff may be too conservative for some species.

We found a positive relationship between urbanization (imperviousness) and wildlife seroprevalence (intercept: −1.665 ± 0.48 SE, urbanization slope 0.039 ± 0.02 SE, *p* = 0.031, Fig. 2a, b). However, antibody detections were highest (80%, *N* = 5) at one of the least urban sites, which is a highly visited state park (average imperviousness 1.5%), and more closely matched seroprevalence at our more urbanized sites (50%, *N* = 10 and 33%, *N* = 12). Human visitation at the state park was similar to human activity in urbanized sites (Fig. 2c), and we found a positive relationship between human presence and seroprevalence of SARS-CoV-2 (univariate linear mixed model with species as a random effect; intercept: -1.132 ± 0.36 SE, human presence coeff: 0.705 ± 0.35 SE, *p* = 0.044; Fig. 2c, Supplementary Fig. 1). This relationship was maintained across different neutralization cutoffs (40–65%; Supplementary Tables 4 and 5).

### Whole genome sequencing from nasal and oropharyngeal swabs
SARS-CoV-2 sequences were obtained for 12 of the 23 RNA positive samples (Fig. 3, Supplementary Figs. 2–9, Supplementary Tables 2 and 3) and Pango lineages were determined for nine total individuals (39.1%; Supplementary Table 2). The SARS-CoV-2 sequence from an opossum trapped in 2022 was assigned to BA.2.10.1 (opossum, *N* = 1) and shared defining mutations in *ORF1a/b*, *S*, *E*, and *M* genes found in the BA.2 Omicron lineage (Fig. 3b and Supplementary Fig. 2 and 11). All eight sequences collected in 2023 were assigned to the XBB* Pango lineages (Supplementary Figs. 3–9, Supplementary Table 2). These lineages were circulating among humans in Virginia during the time of collection (Fig. 3a). The lineages of the sequences obtained from wildlife included XBB (deer mouse, *N* = 1; Supplementary Fig. 3), XBB.1.5 (raccoon, *N* = 1; Supplementary Fig. 4), XBB.1.5.10 (opossum *N* = 1; Supplementary Fig. 5), XBB.1.16 (Eastern cottontail, *N* = 1; Supplementary Fig. 6), XBB.1.5.45 (groundhog, *N* = 1; Supplementary Fig. 7), EG.5.1.1 (deer mouse, *N* = 2; Supplementary Fig. 8), and JD.1 (deer mouse, *N* = 1; Supplementary Fig. 9). The remaining three sequences included two from Eastern red bats and one from a deer mouse, which were generated using amplicon sequencing of the S gene. These matched the SARS-CoV-2 reference sequence (Supplementary Table 3) with 95–100% identity but were at insufficient length for phylogenetic analysis and lineage assignment.

The relative similarity of the sequences obtained from wild animals compared to the closest sequences obtained from humans (Supplementary Table 2) suggest recent introductions of these SARS-CoV-2 lineages into wildlife with at least seven independent introductions of SARS-CoV-2 over a several month period. Interestingly, two sequences from the same lineage (EG.5.1.1) were collected from deer mice on the same day and location (site = PP; Fig. 3 and Supplementary Fig. 8). These two sequences clustered together along with a sequence collected from an infected human in Vienna, Virginia (~330 km

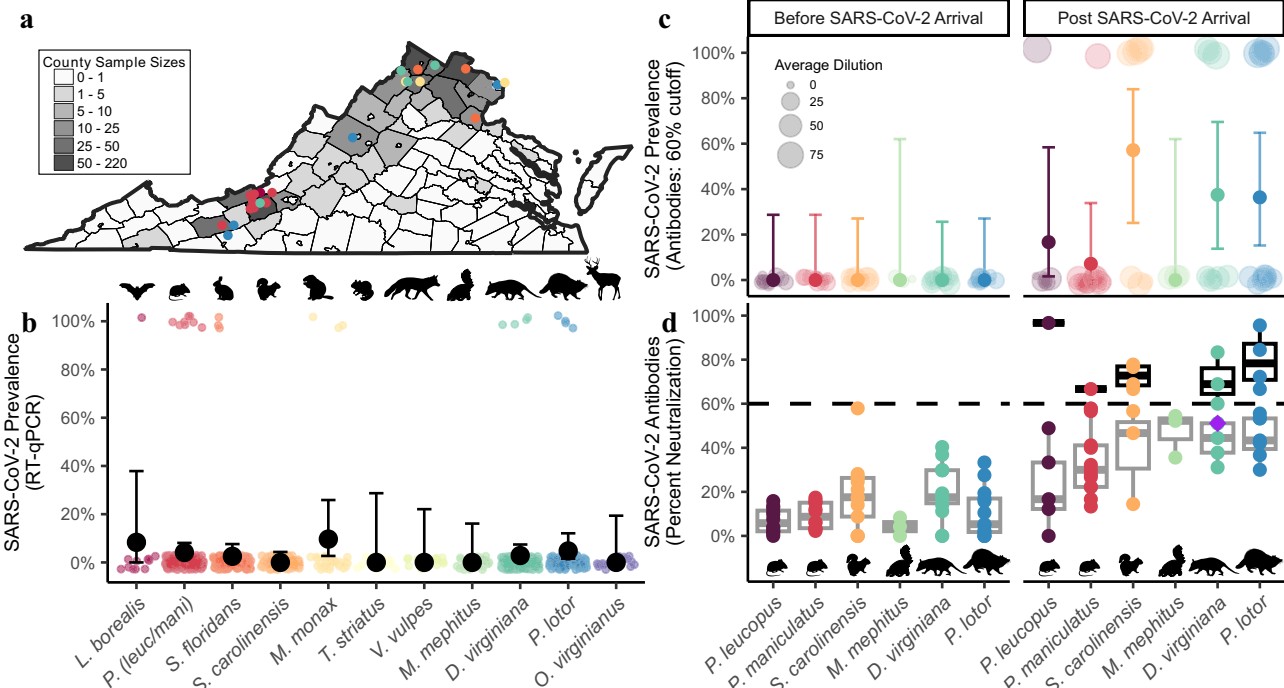

**Fig. 1 | SARS-CoV-2 RNA and neutralizing antibody prevalence in wildlife communities. a** Represents counties where swabs were collected (both wildlife rehabilitation centers and study sites). Gray scale indicates sample sizes. Circles represent SARS-CoV-2 positive samples and color indicates species in (**b**). Prevalence of (**b**) RT-qPCR (quantitative reverse transcription polymerase chain reaction) detections collected between May 2022 and September 2023 (*n* = 757). Detections include species with >10 individuals sampled, and black points indicate prevalence (percentage of individuals with positive detections) and 95% confidence intervals (See Supplementary Table 1 for full dataset that includes all 23 species, *n* = 789). **c** Shows seroprevalence measured using plaque reduction neutralization tests (PRNT) for species sampled both prior to SARS-CoV-2 arrival in 2020 (*n* = 49) and samples collected in our study between June and July 2022 (*n* = 67, Supplementary Data 2). The size of each point in (**c**) is in relation to the percent neutralization. Samples with a 60% or greater percent neutralization were considered positive samples. Solid color points represent seroprevalence

estimates (percentage of individuals positive for neutralizing antibodies) and error bars represent 95% confidence intervals. **d** Represents percent neutralization values of serum samples to SARS-CoV-2 sampled both prior (*n* = 49) and post (*n* = 67) SARS-CoV-2 arrival. The black dotted horizontal line represents a 60% neutralization cut-off. The purple diamond represents neutralizing antibody titers for a Virginia opossum who was found conclusively RT-qPCR positive one month prior. Box plots show the median (center line) and interquartile range (25th–75th percentile of the data) and whiskers indicate range of data 1.5 times the interquartile below and above the 25th and 75th percentile, respectively. Gray boxes represent samples with less than a 60% neutralization value, and black boxes represent samples with greater than a 60% neutralization value. All data points represent samples taken from individual animals which serve as biological replicates. For each PRNT we performed three technical replicates, which were averaged for each individual. Organism silhouettes were sourced from PhyloPic[82].

distance) three days prior (Fig. 3 and Supplementary Fig. 8). This clustering suggests the deer mice might have been exposed to the same source of the virus or could represent a mouse-to-mouse transmission event (Fig. 3c). Sequence from a third mouse collected from the same day and site (site = PP) aligned with another lineage (JD.1, Fig. 3b and Supplementary Table 2), and suggests a separate human-to-mouse exposure (Fig. 3c). At a nearby site (site = NRT; located ~ 52 km away) sequences produced for two additional individuals (one deer mouse and one raccoon, Fig. 3b and Supplementary Table 2) were also from separate lineages (XBB and XBB.1.5, respectively), further suggesting independent human-to-animal introductions (Fig. 3c).

All sequences from wildlife were aligned to their closest related human sequence based on phylogeny, and then assessed for any unique amino acid substitutions. Whole genome sequencing (WGS) of the SARS-CoV-2 variant collected from a positive opossum in July 2022 revealed several mutations shared within the Omicron clade as well as a unique amino acid substitution not previously identified in other SARS-CoV-2 virus samples collected from humans at the time of sampling (Table 1).

The Omicron sub-lineage, BA.2.10.1 (Fig. 3), includes the G[798]D mutation found in the Spike (S) protein. The nearest neighbor (EPI_ISL_14334179), which was collected in New York state 11 days prior to our sampling, contains all the nucleotide and amino acid mismatches found in the opossum sequence, compared to the reference

sequence from Wuhan, China, except for A22974T (S:E[471]V) which appears to be unique to the opossum. Only one mutation was found among the eight other SARS-CoV-2 sequences from 2023 as compared to closely related human sequences. However, this mutation (S:H146Q), in a positive detection from a groundhog, is a common mutation found in other Pango XBB* lineages circulating in 2023[20].

## Unique SARS-CoV-2 mutations collected from wildlife

We investigated the unique E[471]V mutation detected in the opossum sequence from 2022, which was located in the receptor-binding motif (RBM, residues 437 to 508) within the receptor-binding domain (RBD, residues 319–541) of the S protein. The RBD is necessary for binding the viral S protein to ACE2 in both human and animal cells and is therapeutically targeted by neutralizing monoclonal antibodies[21]. We conducted in silico studies to examine the local impact of the E[471]V and G[798]D mutations on the S protein. We employed molecular modeling and molecular mechanics/generalized borne surface area (MM/GBSA) free energy calculations to predict the free energy of binding interaction ($\Delta G_{bind}$). Our focus was on examining the local impact of E[471]V and G[798]D on the favorability of S protein-hACE2 interaction and the modifications in the S2 site as a hypothesis for the experimentally observed traits found in SARS-CoV-2 wildlife reservoir. Here, the open conformation structure of the S protein trimer from the glycosylated BA.2 Omicron variant was used for residue variant mapping [PDB

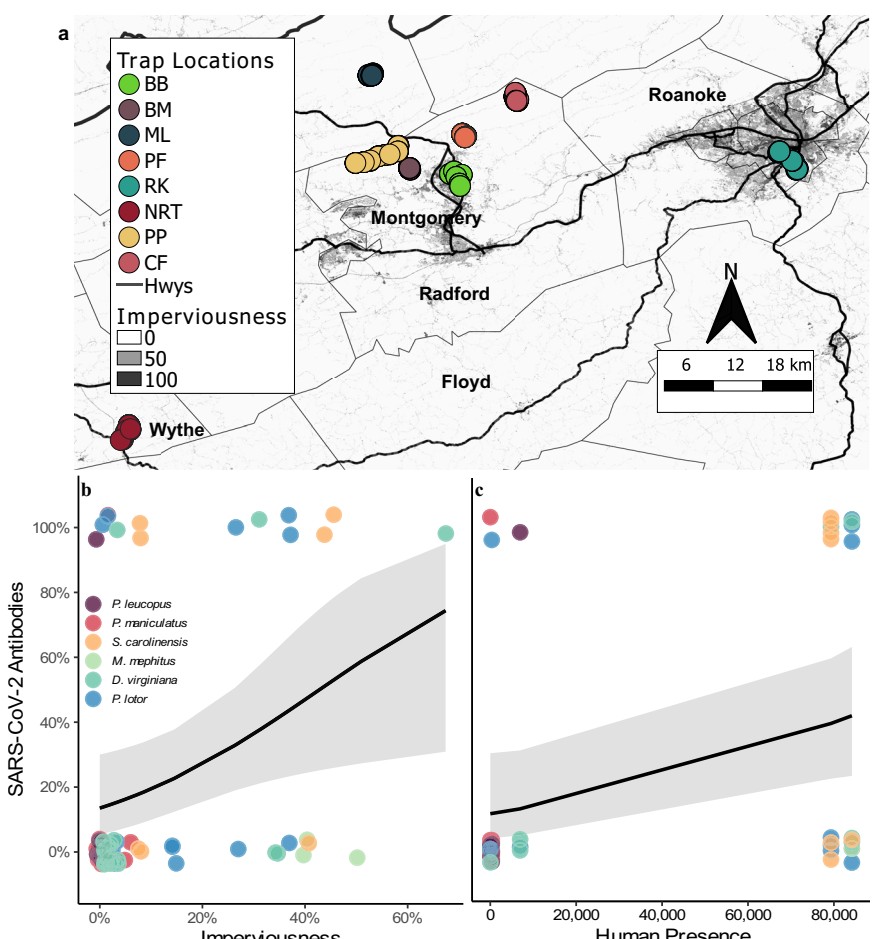

**Fig. 2 | Examination of SARS-CoV-2 exposure at the human-wildlife interface.** **a** Shows imperviousness with darker shades representing more urban areas. Circles represent trap locations (*n* = 856), and sites (*n* = 8) are indicated by different color circles (see Supplementary Methods for trap locations). Sites PF and NRT were sampled in both 2022 and 2023, sites BB, BM, ML, and RK were only sampled in 2022, and sites PP and CF were only sampled in 2023. Serology samples were collected at all study sites in 2022 except PF. **b** Shows the positive relationship (intercept: −1.665 ± 0.48 SE, urbanization slope 0.039 ± 0.02 SE, *p* = 0.031) between urbanization (imperviousness) and seroprevalence collected from animals (*n* = 67)

at five different sites in VA, USA. Panel (**c**) shows the effects of average monthly human presence (population density collected from the 2020 U.S. Census within the trapping area of each site (Blacksburg) or use estimates from trail counters or landowners) on seroprevalence (intercept: -1.132 ± 0.36 SE, human presence coeff: 0.705 ± 0.35 SE, *p* = 0.044, *n* = 49). Black lines indicate model estimates from generalized linear mixed models and the gray ribbons (shaded areas around the line) represent 95% confidence intervals. Color circles in (**b**) and (**c**) indicate species sampled.

7XO8][22]. Because the crystal structure shows the trimer bound to the human ACE2 (hACE2) receptor, it is possible to investigate the local impact of mutations in the RBD on interactions between SARS-CoV-2 S protein and the human ACE2, as a prediction for mechanistic impact on S protein-ACE2 interaction favorability as previously reported and validated experimentally[23] (Fig. 4). Our free energy studies predict that $\Delta G_{bind}$ between the S:E[471]V-carrying mutation and hACE2 was improved relative to that of BA.2 with hACE2 (-76.5 and -60.2 kcal/mol for E[471]V vs. BA.2, respectively). Large-scale low-mode conformational sampling was performed on the RBD of BA.2 and E[471]V Spike (residues 329-531) with hACE2 to probe $\Delta G_{bind}$ on multiple potential conformations[24,25]. Consistent with the results observed in full-length Spike-hACE2 interactions, the mean $\Delta G_{bind}$ for the top five most favorable E[471]V$_{RBD}$ conformations (-151.7 ± 19.2 kcal/mol) was more favorable than the mean for BA.2$_{RBD}$ conformations (-125.4 ± 4.1 kcal/mol, $p \leq 0.05$). The sampled conformations of E[471]V$_{RBD}$ exhibited a larger range of $\Delta G_{bind}$ values relative to BA.2$_{RBD}$, suggesting a minor variation in the nature of interactions in the RBD due to the E[471]V mutation. Residue 471 is predicted to be part of a flexible, hydrophobic loop that interacts with the N-terminal domain of ACE2 (Fig. 4). Several point mutations (A[475]V, S[477]G, V[483]A, F[490]L) within this loop region have been experimentally

shown to improve S protein – hACE2 binding affinity and increased resistance to neutralizing antibodies[26,27]. Our analysis using normalized Kyte-Doolittle hydropathy scores and surface maps show a more hydrophobic interaction surface at not just position 471, but also among flanking residues 469-474 (0.8) compared to BA.2 [-2.6][28]. The increased favorability of the E[471]V mutation for hACE2 interaction aligns with previously reported trends of residues in the RBD influencing stabilization and enhanced affinity[29]. Any differences may be attributed to entropic effects, emphasizing the potential enthalpic contribution of this mutation to improve hACE2 interaction at the RBD interface.

We mapped the second missense mutation, G[798]D, in the S2 subunit of the S protein (residues 686 to 1,273) and within the fusion peptide domain (residues 788–806) near the N[801] glycosylation site. When glycosylated, the N[801] site significantly enhances viral entry, with very low mutation rates observed in adjacent residues (e.g., within 2–3 residues from position 801)[30]. Interestingly, D[798] creates a small, charged pocket on this solvent-accessible S2 loop, decreasing the probability of glycosylation at N[801] (0.48) compared to BA.2 [0.61][31]. It is hypothesized that G[798]D could impact structural stability and membrane interaction[32] (Fig. 4

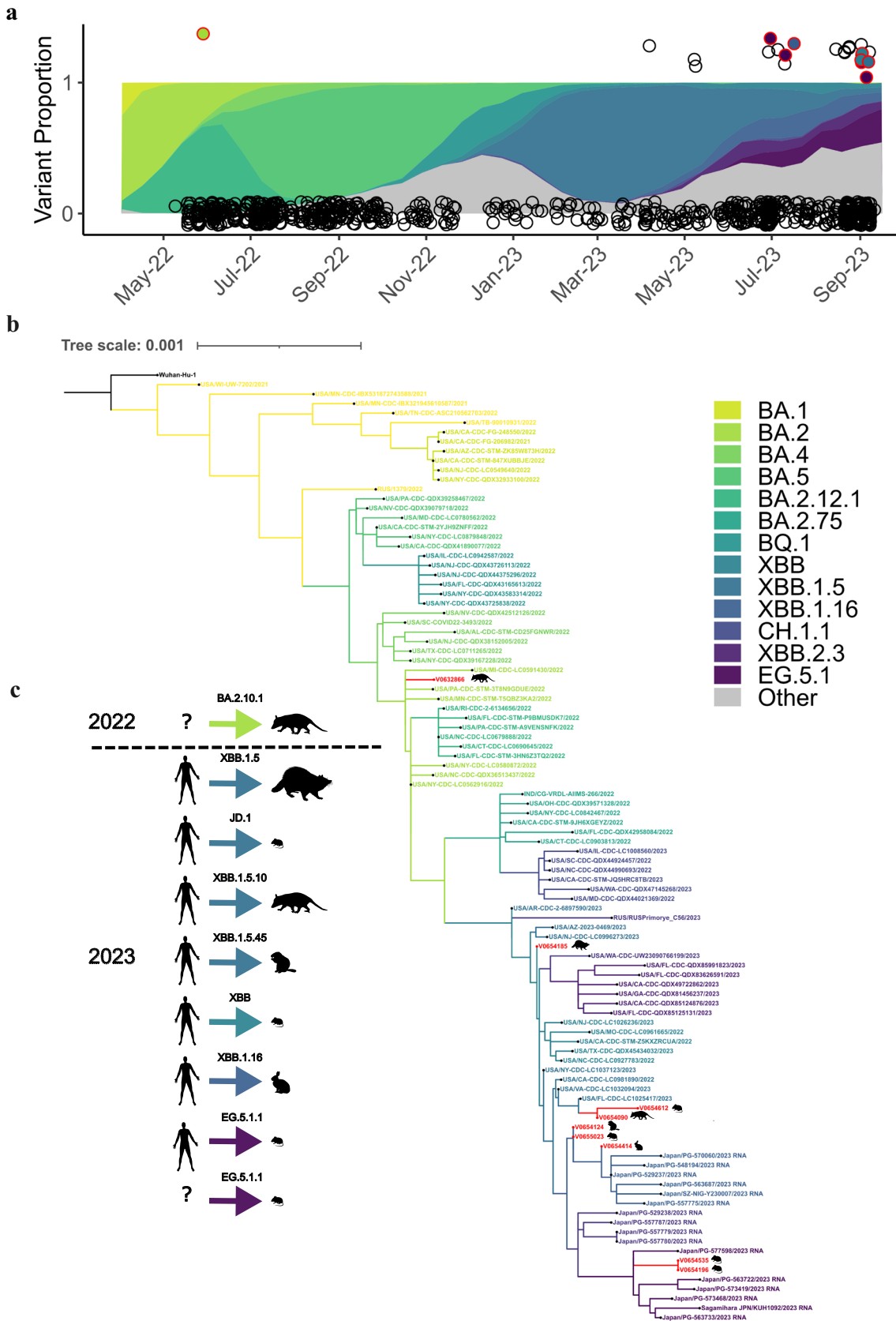

and Supplementary Fig. 10) and have a negative impact on Spike maturation and infectivity. While this approach limits the investigation of these mutations and their role in the dynamic conformational states of the S protein, it provides a foundation for understanding the mechanistic impact and connection of structure to experimental observables.

**Fig. 3 | SARS-CoV-2 collected from wildlife in comparison to human samples.**
**a** Color ribbons show the proportion of each lineage circulating in humans in sampling region at the time of data collection[83]. Circles represent each individual animal sampled ($n = 789$). A positive sample has a value of 1 and a negative sample has a value of 0. Circles outlined in red represent a positive sample with lineage assignment ($n = 9$, Supplementary Figs. 2–9; Table 3). The fill color within each red circle represents the lineage it was assigned. **b** A phylogenetic tree was generated using maximum likelihood analysis and GTR + G4 substitution models to visualize the relatedness of the nine whole genome sequences obtained from wild animals ($n = 9$) to SARS-CoV-2 sequences derived from human hosts ($n = 90$) representing different PANGO lineages. Additional phylogenetic analyses show the placement of the SARS-CoV-2 sequences from each individual wild animal within individual lineages in more detail (Supplementary Figs. 2–9). **c** Suspected transmission origin for each individual is represented by a human silhouette or a question mark ("?") if the origin is suspected to be either a human-to-animal or animal-to-animal transmission. Inferences about transmission were determined by phylogenetic analysis using related human host sequences and our sequences collected from wild animals as shown in Supplementary Figs. 2–9 and in Supplementary Table 2. We also considered any unique mutations not described from a human host, as those mutations may have come from the originating hosts' virus. The arrow color indicates the lineage assignment and silhouettes indicate the species from which the sample was collected. Across all panels color indicates the same lineage of SARS-CoV-2.

## Discussion

Our combined results suggest that a broad diversity of mammal species have been exposed to SARS-CoV-2 in the wild (Supplementary Table 6). While species like the white-tailed deer have been shown to be important hosts for SARS-CoV-2, our results highlight that evaluating the importance of each species in the context of a broader community of hosts will be critical for controlling future zoonotic disease risk[33].

For some species, SARS-CoV-2 RNA detections and seroprevalence aligned well with both predicted susceptibilities based on ACE2 modeling and experimental infection studies (e.g., whether the species seroconverted, was capable of viral shedding, or transmitting; Supplementary Table 7). Most notable is the deer mouse, which has been predicted to have high susceptibility[34], shown experimentally to be capable of mouse-to-mouse transmission[12,13], and produces antibodies in the lab[12,13,35]. We had the highest number of positive RNA detections ($N = 8$) in this species, as well as detected neutralizing antibodies (Fig. 1, Supplementary Table 1 and 6). Raccoons have also been shown experimentally to be capable of seroconverting following SARS-CoV-2 exposure[14,35], and we found high seroprevalence (63.6%) in this species. However, experimental infections did not find evidence of viral shedding or transmission in raccoons when infected with the USA-WA1/2020[14] or the WA1/2020WY96 SARS-CoV-2 isolates[35], whereas we had four positive RT-qPCR detections in this species (Supplementary Table 1). We also had positive RNA detections in cottontail rabbits ($N = 3$), which, when experimentally infected in the lab, also did not produce any clinical signs of infection, or appear to be capable of viral shedding when infected with the WA1/2020WY96 isolate[35]. Several factors may contribute to the apparent discordance between experimental infections and positive detections in the wild. First, experimental infection studies with these species used earlier SARS-CoV-2 variants and most positive detections in this study occurred recently (July-September 2023, Fig. 3a) with newly emerged lineages (e.g., XBB). The rapid evolutionary change in SARS-CoV-2 infectivity, virulence, and immune escape abilities may have changed species competence for SARS-CoV-2. Supporting this, recent research has shown that Omicron's (B.1.1.529) ACE2 binding affinity is greater in a number of wildlife species compared to humans, including white-footed mice, red fox, two marmot species, and five marsupial species[36]. Another possibility is that some RNA detections do not reflect active infections, but rather recent exposure to another infected animal. Other species with positive detections (groundhog, Virginia opossum, and Eastern red bat) have not currently been evaluated in laboratory experiments (Supplementary Table 7). Experimental infection studies and updated molecular modeling with new viral lineages will be essential for assessing the current risk to other mammal species.

Many of the species in this study are considered human peri-domestics, have broad ranges across North America, and live in and around human settlements. In addition, some species have been introduced into Europe (gray squirrels and raccoons) and Asia (raccoons) and have close relatives across the globe. Deer mice are of particular importance given that they are known to be reservoirs for other pulmonary viruses[37], and are in close direct and indirect contact with humans. It is unknown whether a sylvatic cycle has been established in any of the species we evaluated; however, their close connection to humans and broad spatial distribution means they are likely to experience continuous exposure to a diversity of SARS-CoV-2 lineages in the future.

We found support for a relationship between human presence and seroprevalence, suggesting that areas with high human activity may serve as potential hotspots for cross-species transmission. This suggests that wildlife in areas with more human activity likely have higher risk of exposure to SARS-CoV-2. Hence, recreational and highly urbanized sites may be important points of contact between humans and wildlife and could be targeted for surveillance and control. Humans and wildlife rarely come into direct contact, but numerous indirect links likely exist. Wastewater has been proposed as a potential source for indirect exposure to SARS-CoV-2[38], however, in rural areas where septic tanks are a dominant form of wastewater management, this is unlikely to be the only source. Instead, other forms of human waste, like trash receptacles, may be important sources of indirect SARS-CoV-2 exposure in wildlife[14,39]. Urban wildlife are regularly exposed to human refuse and have developed positive associations with discarded food, which could serve as a bridge for transmission between humans, companion animals, and wild species[40–42].

We obtained eight RNA sequences in 2023 from multiple Pango lineages, likely indicating multiple introductions of the virus from humans into wildlife (Supplementary Table 2). This highlights that wildlife are continually exposed to a wide diversity of SARS-CoV-2 lineages, as has been observed in white-tailed deer[5,18]. We can further assume that these are recent introductions, and not due to an established animal host sylvatic cycle, as they are close matches (99.1–100.0%) to published human sequences circulating at the same time of collection. It is important to note that in two instances, multiple animals tested positive at the same site during the same collection period (~4 days), and animal-to-animal transmission following human introduction cannot be ruled out. It is difficult to fully resolve whether transmission occurred via human-animal or animal-to-animal in the summer of 2023 because of a lack of human collected samples from that time, as the reporting of human samples in 2023 had significantly decreased compared to previous years. Hence, there is a much smaller sample size of known circulating human variants at a fine resolution.

**Table 1 | Summary of mutations identified in the Virginia opossum (*Didelphis virginianus*)**

| SNP | Gene | Mutation |
|---|---|---|
| C7749T | *ORF1ab/NSP3* | Missense T$^{2495}$I |
| T16342C | *ORFab/Helicase* | Missense S$^{5390}$P |
| A20304G | *ORFab/EndoRNAse* | Synonymous |
| A22974T | *S* | [1]Missense E$^{471}$V |
| G23955A | *S* | [2]Missense G$^{798}$D |

[1,2]Figure 4a, Supplementary Figs. 10 and 11. *Italics* indicate the SARS-CoV-2 gene.

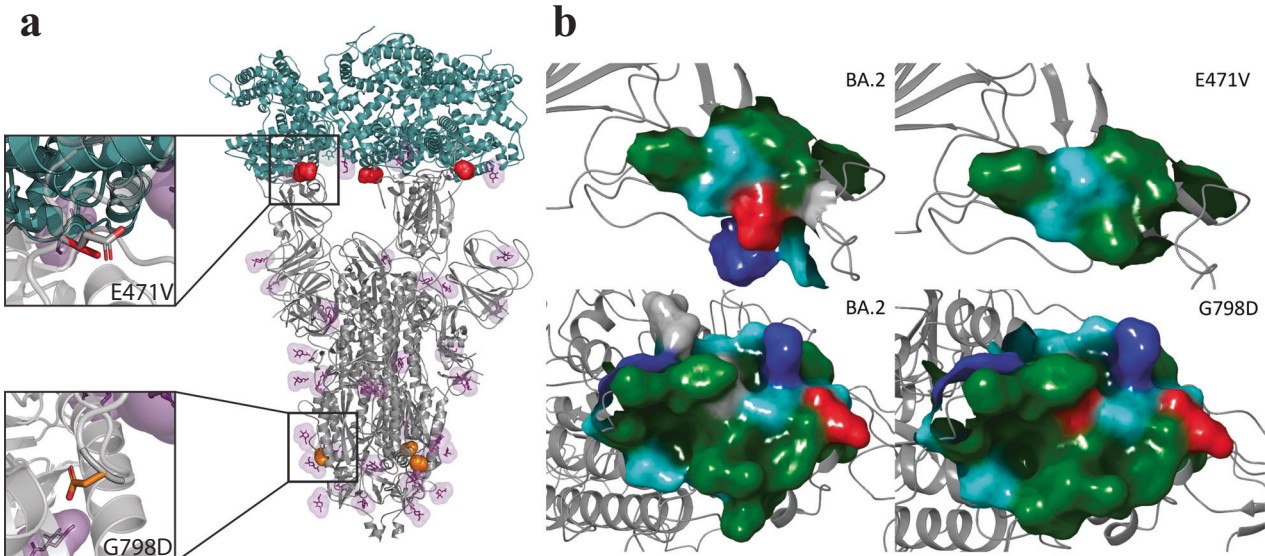

**Fig. 4 | Molecular modeling of the unique S mutation of SARS-CoV-2 collected from a Virginia opossum. a** Representation of the structure of the BA.2 S protein (gray, PDB: 7XO8) in its open conformation bound to the human ACE2 (teal). Residues 471 (red) and 798 (orange) are shown as spheres. Glycans are displayed as sticks colored purple. Top inset: Overlay of E$^{471}$V (red) and BA.2 (gray). Bottom inset: Overlay of G$^{798}$D (orange) and BA.2 (gray). **b** Surface map of the BA.2 S protein (left) and the region surrounding E$^{471}$V and D$^{798}$G (right) residues. Residue side chain properties are colored: green for hydrophobic, blue for positively charged, red for negatively charged, teal for polar uncharged, and gray for neutral.

Determining how wildlife are being infected (pathways such as human refuse, wastewater, contact with infected pets, etc.) is a critical next step in disease control and management. It will be important to continue to sequence variants from wildlife as well as humans to assess if SARS-CoV-2 is adapting to new wildlife hosts, if a sylvatic cycle develops, and whether there is a risk of transmission back to humans.

Analysis of sequences collected from an opossum in 2022 revealed a novel mutation that has not been described in any other variant, which might suggest animal-to-animal transmission or possible adaptation within the opossum. All other mutations were found circulating in humans in other regions. Some of these mutations likely increase binding affinity to the hACE2 receptor or confer some antibody resistance compared to ancestral lineages. Specifically, SARS-CoV-2 collected from the opossum had mutations in the RBM of the RBD of the S protein. The previously uncharacterized amino acid change is predicted to improve S-hACE2 binding compared to Omicron BA.2, potentially providing a fitness advantage by either increasing the affinity of S for the ACE2 receptor or, alternatively, by evading the neutralizing activity of antibodies. Whether these mutations developed in humans not captured in surveillance at the time or in wildlife communities and were transmitted back to humans, which has been previously suggested during the emergence of the Omicron variant[5,7], remains unknown (Fig. 3). Future work using molecular dynamics (MD) simulations to evaluate the impact of these mutations on structural stability, membrane interaction, and conformational states will help contribute to our understanding of evolutionary pressures and cross-species transmission.

While single gene amplification (*S, N, or E*) below our cutoff did not meet our criteria for assigning a positive (*i.e.*, at least two SARS-CoV-2 genes need to amplify below the cutoff for a reference sample to be reported as positive based on our EUA-FDA and CLIA protocols), the analytical specificity (cross-reactivity) of the RT-qPCR (99-100% single gene specificity) suggests that many of these may be positive detections with low levels of RNA[43]. Single-gene positive detections overlapped with all six species confirmed in this study (Supplementary Table 1), but also encompassed an additional seven species (Supplementary Tables 1 and Supplementary Data 1), including the American beaver (*Castor canadensis*; 1/2), bobcat (*Lynx rufus*; 2/3), American black bear (*Ursus americanus*; 1/7), red fox (*Vulpes vulpes*; 2/17), white-tailed deer (3/20), skunk (1/25), and Eastern gray squirrel (4/105). We also obtained sequence data that was a 100% match for SARS-CoV-2 from single gene detections, further suggesting that in some cases, these may indicate true positives (Supplementary Tables 2–3 and Supplementary Data 1). While confirming new species requires a more conservative and confirmatory approach, given the large number of single-gene positives, future studies should carefully consider whether these are unconfirmed detections, particularly in small wildlife species, which may produce lower viral loads than humans or larger mammals like deer. Importantly, classifying unconfirmed samples as negative could substantially alter our understanding of the ecology and community dynamics of SARS-CoV-2 in wildlife going forward. Furthermore, our use of neutralization tests to detect evidence of prior SARS-CoV-2 exposure may have resulted in reduced sensitivity over approaches like ELISA that identify only binding antibodies. However, binding-antibody-based approaches also suffer from false-positivity to other coronaviruses[44], other infectious diseases[45], and even autoimmune diseases[46]. Thus, PRNT increases the confidence that animals tested here were truly exposed to SARS-CoV-2 and not to another related viruses.

Our results have greatly expanded the known range of hosts exposed to SARS-CoV-2 in the wild, which now includes nine species (six from this study) that have been documented with SARS-CoV-2. Several of the species that tested positive (mice, rabbits, and opossums) have traits that may make them more suitable for establishing a SARS-CoV-2 wildlife reservoir than previously described species. These include fast-paced life history strategies whereby individuals reproduce frequently at a young age, and in some cases are known reservoirs for other respiratory viruses[47,48]. Continued broad surveillance and more detailed ecological research will be needed to fully determine the role of wildlife communities in SARS-CoV-2 transmission and evolution.

## Methods
### Study Sites
We collected nasopharyngeal or oropharyngeal samples from wildlife across 43 counties in Virginia and Washington D.C., U.S.A. We collected

samples from three wildlife rehabilitation centers in Boyce, Roanoke, and Waynesboro, VA. In addition, we actively captured wildlife from eight sites between 2022 and 2023 that spanned a rural to urban gradient in Giles, Montgomery, Roanoke, and Wythe counties (Fig. 2a, Supplementary Table 8, and see Supplemental Methods for site descriptions.). At six sites we actively trapped wildlife using live-traps from 9-May through 1-July 2022 and at four sites (two sites were the same as 2022) we trapped from 27-June through 11-September 2023.

## Trapping and processing

We trapped each of six sites for a 2–4 day session between 9 May and 1 July 2022 (two sites were trapped for two sessions with at least 27 days between them). Additionally, we trapped four sites for a 4 day session between 27-June and 11-September 2023 (Fig. 2a). We used three different sized live traps to capture animals: Tomahawk non-folding traps (Tomahawk Live Trap Co., Tomahawk, WI, U.S.A; 46 × 15 x 15 cm, model 103.5), Tomahawk folding traps (66 × 23 x 23 cm, model 205 or 81 × 25 x 31, model 207), and Sherman folding traps for smaller animals (Sherman Traps, Tallahassee, FL, U.S.A; 8 × 9 x 23 cm, model LFG). We set and baited traps in the evening just before dark and checked them the following morning before 10:00 AM to ensure animals did not get overheated.

We processed all animals in the area where they were trapped. We anesthetized larger animals in either a bucket chamber[49] or in a 50 L clear box chamber (58 × 38 x 56 cm, Supplementary Fig. 12). We used a tec-4 funnel fill with a cage mount manifold vaporizer (Ohmeda, West Yorkshire, UK) connected to a portable medical oxygen cylinder (size E) with a built-in regulator (Walk-O2-Bout+™, Airgas Healthcare, Radnor, PA). We used a 2 L/min oxygen mix to 3–4% isoflurane in the chambers. Smaller animals were placed in a small plastic canister and anesthetized with 0.5–1.2 mL isoflurane (dosage depended on species and body size), placed onto a cotton ball held within a perforated canister to prevent direct contact with the animal (Supplementary Fig. 12). Once anesthetized, we removed animals from the chamber, and then masked animals (except for mice; Supplementary Fig. 12) with a dose of 2 L/min oxygen and 2–3% isoflurane.

We marked individuals with aluminum ear tags in each ear (National Band and Tag Company, model 1005-1 or 1005-3 depending on body size). We collected morphometric, sex and reproductive status from each animal. For RT-qPCR detections, we swabbed each individual with a polyester swab (25-800 1PD swab for larger mammals and 25–1000 1PD swab for smaller animals e.g., juveniles, skunks, squirrels, bats, shrews, and mice; Puritan Medical Products, Guilford, ME, U.S.A). Given the small swab size compared to humans and larger mammals and the reduction in material that could be collected from these swabs, we collected two swabs per animal after July 2023. We used an oropharyngeal swab for animals that were too small for nasopharyngeal swabbing (mice, shrews, bats, etc.). All collection swabs were placed in tubes with transport media[43] and kept on ice until they were transported back to the laboratory following field work each day. Transport media consisted of a 1:1 volume of Dulbecco's Modified Eagle's Medium (DMEM) containing low glucose (1 g/l), sodium pyruvate and L-glutamine (Corning) media and 2x DNA/RNA Shield™ (Zymo Research), which can preserve samples up to 10 days at room temperature[43]. Field collected samples were all stored in the refrigerator at 4 °C and processed within five days following field collection. We collected blood for antibody screening from the submandibular vein for mice and from the nail quick of one of the rear toenails for all other species. Blood was stored in microvette 500 z-gel tubes for later processing. We put all blood samples into a cooler immediately following field collection and stored at 4 °C. Three wildlife rehabilitation centers in Virginia provided wildlife oral and nasal swabs following methods described above. We analyzed all samples from rehabilitation centers within 45 days of collection. Protocols were approved under Virginia Tech IACUC protocol #22-061.

## Personal protective equipment and contamination control

All personnel collecting and working with the samples wore fit-tested N95 respirators and gloves. We sterilized all equipment between each animal. We also sterilized all traps and equipment between each trapping session, and individuals collecting the samples were tested using the assay described below 1–2 times per week to confirm they were negative throughout the duration of sampling. During fall of 2023, when most positive samples were collected, all personnel were tested daily.

## RNA Extraction and detection of SARS-CoV-2 viral RNA

The RT-qPCR based test we used for the identification of SARS-CoV-2 RNA in wildlife has been thoroughly described in Ref[43]. This SARS-CoV-2 assay, authorized for use in humans by the FDA during the pandemic and adhering to all regulatory standards, detects three targets including the nucleocapsid (N), envelope (E), and spike (S) genes of the SARS-CoV-2 virus, has high analytical specificity, and across all three targets has a 100% probability of detection at 10.0 copies per 10 μL.

We processed swab-containing media shortly after collection and total RNA was purified using 96-well spin columns, assessed for quality control, and subjected to synthesis and amplification using the Power SYBR™ Green RNA-to-CT™ 1-Step kit (Applied Biosystems). The cytochrome c (Cytochrome C Oxidase Subunit 1; Cox1) housekeeping gene, for which species-specific primers were designed (Supplementary Data 3), and HRPP30 housekeeping gene for human samples was used as a control for whether sufficient sample was collected from each individual. RT-qPCR reactions were performed in a CFX384 Touch Real-Time PCR detection system (Bio-Rad), no template controls, and standard curves on each plate as previously described[43]. Samples were reported conclusively positive when two or more SARS-CoV-2 targets (N, E, and S genes) and the housekeeping gene (Supplementary Data 1) amplified below the threshold established by corresponding standard curves[43], which is similar to the criteria established for the Applied Biosystems TaqPath COVID-19 used for confirmation in other wildlife studies[7]. The use of multiple targets is particularly relevant when considering genetic variations of SARS-CoV-2 among circulating variants for which the sole amplification of a single gene could result in a false negative result if the mutation was in a region of the genome assessed by the test. Thus, a molecular diagnostic test developed to detect multiple genetic targets of SARS-CoV-2 is likely less susceptible to novel genetic variation. For the positive samples from the opossum collected in 2022, the remaining sample was sent to the USDA National Veterinary Services Laboratories (NVSL) in Ames, Iowa for confirmation of results. However, while insufficient sample remained for NVSL to conduct confirmatory testing, raw sequence data from this positive was independently analyzed and confirmed to be from an opossum.

The Virginia Division of Consolidated Lab Services (DCLS) confirmed the performance of the developed assay using a comparator assay that followed the CDC EUA IFU (CDC DOC 006-0099 rev.03) "CDC 2019-Novel Coronavirus (2019-nCoV) Real-Time RT-PCR Diagnostic Panel" that includes the same probes and N primers. Results of all blindly tested samples were incorporated into the Emergency Use Authorization which was approved by the Food and Drug Administration. To date, this assay has been used for sample analysis by the Virginia Department of Health in over 230,000 tests in Southwest Virginia. Lastly, the Molecular Diagnostics Laboratory that tested the samples is CLIA certified and participates in a proficiency testing program from the American Proficiency Institute.

## Serology data

We obtained additional samples (Supplementary Data 2) to compare seropositivity prior to the emergence of SARS-CoV-2 to those samples we collected in 2022[50]. The samples were transferred to gold-top microtainer tubes (BD, Franklin Lakes, NJ, USA) and serum was separated via centrifugation at 5000 x g for 10 min and transferred to

1.7 mL tubes. Serum samples were then heat inactivated at 55 °C for 1 h. Plaque reduction neutralization assays (PRNTs) were performed similarly to prior reports[51,52]. Briefly, samples were diluted 1:10 in RPMI-1640 with 10 mM HEPES and 2% FetalPure bovine serum (Genesee Scientific 25-525H). Diluted serum samples were mixed in a 1:1 v/v ratio with a solution containing 1300 plaque-forming units per mL (PFU/mL) of SARS-CoV-2 Delta virus strain USA-GNL-1205/2021 (a generous gift from the World Reference Center for Emerging Viruses and Arboviruses), or ~40 PFU's per well. Our serum samples were collected in June and July of 2022, following the Delta wave in fall of 2021, and given the life expectancy of the animal species (Supplementary Table 9) we were sampling, we assumed Delta would have been the most likely recent variant they were exposed to[53]. Each sample was tested in triplicate. The mixture was then incubated at 37 °C for 1 h, and then 50 μL of the virus-serum mixture was used to inoculate wells in a confluent 24-well plate of VeroE6 hACE2-TMPRSS2 cells and again incubated at 37 °C for 1 h. After the 1-h adsorption period, we added 500 μL of overlay media to each well and incubated the plates at 37 °C, as previously described[54]. After 2 days, the plates were fixed with 10% formalin and stained with crystal violet. The level of neutralization is based on the reduction of plaque counts compared to virus mixed with diluent. Based on previously published sero-surveillance studies[55–58], we used the following testing criteria:

1. Greater than or equal to 90% Neutralization at a 1:20 Dilution– Strong Positive
2. Greater than or equal to 60% Neutralization at a 1:20 Dilution– Weak Positive
3. Less than 60% Neutralization at 1:20 Dilution–Negative

Any individual with a 60% neutralization or higher was counted as positive in analyses analyzing the effects of urbanization and human presence on SARS-CoV-2 exposure.

### Urbanization and human presence variables

We obtained estimates of urban imperviousness from the National Land Cover Database 2019[59,60]. We obtained estimates of population density from 2020 census at the 1 km spatial scale[61]. We created buffers of variable sizes based on species home range size to calculate the mean estimates of imperviousness and population density for each individual capture location (Supplementary Table 10).

We obtained estimates of human presence at the five sites where we collected serology data (Supplementary Table 8), and all visitation data was standardized to number of visits per month. New River Trails State Park and Roanoke Parks and Recreation provided monthly human visitation estimates between March and August 2022. Mountain Lake Biological Research Station provided estimates for how many people used the facilities at the station between March and August 2022; they averaged about 50 people at the station per month with each person staying 7 days on average. Brush Mountain recreational area was closed to the public from March through August 2022, however, there was some trail maintenance work and the occasional trespasser, so we assumed ~10 people used the area per month. For the town of Blacksburg, we used population estimates from the 2020 census (TIGER/Line, U.S. Census Bureau, 2020 Blocks). We created a minimum convex polygon (MCP) in the adehabitatHR package[62] in program R v 4.1.1[63] around the different trapping locations. We then added a 1 km buffer area around each census assuming both animals and residents move in and out of the trapping mcp area. We used ArcGIS Pro (ArcGIS Pro 2.9.0, Esri, Redlands, CA, USA) to calculate the population within each mcp area. We used the intersect tool field calculator, and summary statistics to calculate the population for each census block by multiplying the percent of the block within the mcp by the population count. We then summed the population within the mcp to estimate the total population. We then multiplied the population count by 30

(approximately days in each month) to compare the daily estimates of occupancy to the monthly estimates from the other sites.

### Clinical specimen analysis

Human (13,221) and wildlife (789) clinical samples were analyzed for the presence of SARS-CoV-2 by RT-qPCR. We performed whole genome and amplicon sequencing for positive samples. We limited the description of variants in the human population to the window at which wild sample collection took place (May 1, 2022, to September 8, 2023, Fig. 3a and Supplementary Fig. 13).

Human clinical samples were collected as part of ongoing SARS-CoV-2 surveillance efforts in the region and collected under approval through Virginia Tech Institutional Review Board (IRB 20-852) and the Virginia Department of Health (VDH IRB 70046) and consent was obtained for the use of all human samples. The analysis of human clinical specimens identified 4,123 as positive, 8,350 as negative, 162 as inconclusive and 586 as invalid (no housekeeping gene detected), respectively. (Supplementary Fig. 13a). A few samples failed sequencing or were not assigned due to their quality/quantity and low sequence depth. As shown in Supplementary Fig. 13b, to further examine the sequence data from the positive opossum from 2022, various SARS-CoV-2 lineages expanded the window of our study with BA.1 (0.17%), BA.2 (15.85%), BA.2.12.1 (25.80%), BA.4 (14.83%), BA.5 (37.91%), BE (1.57%), BF (1.45%), BG.2 (0.05%), AY.103 (0.02%), not assigned (N/A, 2.24%) being the most prevalent among humans in the region.

### Whole genome sequencing and Sanger amplification

Sanger sequencing was carried out using ARTIC primers (v3/v4.1) *. Samples were treated with ExoSAP-IT™ Express (Applied Biosystems) and submitted to Eurofins Genomics (Louisville, KY) for Sanger sequencing. To amplify the entire SARS-CoV-2 genome of the 2022 Virginia opossum, positive samples were reverse transcribed and amplified using SuperScript IV Reverse Transcriptase (Thermofisher) and ARTIC nCoV-2019 Amplicon Panel V4.1 primers (Integrated DNA technologies). Purified PCR products were barcoded using the plex-Well™ 384 Library Preparation Kit (seqWell, MA) and the pooled library sequenced using a MiSeq System (Illumina, CA) following manufacturer's instructions. SARS-CoV-2 sequencing analysis was accomplished using an established pipeline and optimized workflow for validation that combines tools to assess quality, sequence alignment, variation calling, and variant assignment. We processed lineage assignments using Pangolin software suite (https://cov-lineages.org/)[64].

All remaining positive SARS-CoV-2 detections in 2023 by qRT-PCR were whole genome sequenced using a GridION X5 Sequencer (Oxford Nanopore Technologies, ONT) separately from the 2022 Virginia opossum. Amplifications were performed using the Lunascript RT Supermix (New England BioLabs) and Midnight RT PCR Expansion (Oxford Nanopore Technologies). Barcoding was performed using the Rapid Barcoding Kit 96 (Oxford Nanopore Technologies) before the cDNA pool was purified. The pure cDNA pool was quantified using either a Nanodrop 2000/2000c or Qubit Flex (Thermo Fisher Scientific) before loading into a flow cell (R.9.4.0 or R.9.4.1 Flow Cell Versions; Oxford Nanopore Technologies). Sequencing was performed using a GridION X5 Sequencer. Raw sequence data was basecalled and demultiplexed using ONT's Guppy software, and analysis performed using ONT's EPI2ME Labs software package running the wf-artic analysis protocol using scheme Midnight-ONT/V3 (Oxford Nanopore Technologies and ARTIC Network). The wf-artic analysis protocol utilizes Medaka, Nextflow, Nextclade, and Pangolin to produce a comprehensive analysis report that assesses sequence quality and alignment, as well as variant calling and assignment. Whole genome sequences of all human samples were deposited in the GISAID database (https://www.gisaid.org/) and NCBI SARS-CoV-2 sequence repositories.

## Phylogenetic analysis

Consensus FASTA files containing available sequences from shotgun whole genome sequencing and targeted amplicon sequencing were assembled using DECIPHER software package[65]. These sequences were uploaded into Ultrafast Sample placement on Exiting tRee [UShER; https://genome.ucsc.edu/cgi-bin/hgPhyloPlace/][66,67] for global phylogenetic tree analysis using 12,397,869 SARS-CoV-2 genomes obtained from GISAID, GenBank, COG-UK and CNCB that were available as of 2022-10-03 for the original opossum sequence and 16,284,433 SARS-CoV-2 genomes that were available as of 2023-10-12 for the other eight SARS-CoV-2 sequences obtained from wild animals. Sequences were also uploaded into NextClade [https://clades.nextstrain.org/][68] for clade assignment and mutation calling.

For visualization of these nine consensus whole genome sequences in comparison to Pango lineages, a phylogenetic tree was generated using a randomized subset of 90 SARS-CoV-2 sequences from NCBI Virus[69] that represent 15 different Pango lineages (6 sequences per lineage). The sequence alignment and a Maximum Likelihood (ML) tree was generated using DECIPHER's TreeLine function[65] and viewed using iTOL v6[70].

Each of these SARS-CoV-2 sequences collected from wildlife were further analyzed for similarity to published SARS-CoV-2 sequences obtained from humans within our geographical region. We assembled separate phylogenetic trees for each Pango lineage using human SARS-CoV-2 sequences collected from Virginia and the four states that neighbor Southwest Virginia, including West Virginia, Kentucky, Tennessee, and North Carolina (Supplementary Tables 11 and Supplementary Data 4). To limit the size of some of the trees, we only included sequences collected around the same time as the dates for the wildlife sample collection. For lineages XBB.1.4.45 and JD.1, an insufficient number of sequences were available from these five states so all available sequences from North America were included for generating the phylogenetic trees. These human SARS-CoV-2 sequences were downloaded from GISAID[20] and NCBI Virus. Duplicate sequences were removed prior to tree assembly. The trees were rooted using the NCBI Reference Sequence (NC_045512.2) for SARS-CoV-2. Sequence alignments and ML trees were generated as described above.

We used Nextclade's mutation calling feature to identify novel amino acid substitutions that were unique to our wildlife samples as they were not also found in their closest neighboring sequence.

## Molecular modeling

We carried out computational studies of the structure of the glycosylated Omicron BA.2 variant bound to the human ACE2 receptor [Protein Data Bank 7XO8][22]. We incorporated point mutations in BA.2 using PyMOL (The PyMOL Molecular Graphics System v. 2.4.0, Schrödinger, LLC) to generate a new structure containing both E471V and G798D mutations. We then performed energy minimization using Schrödinger-Maestro (v. 2020.4) and OPLS3e force field[71,72]. We used Schrödinger Bioluminate to carry out molecular mechanics/generalized borne surface area (MM/GBSA) energy calculations[73,74] as well as surface mapping [v. 2020.4][28]. In brief, the S protein and hACE complex underwent a second round of minimization using the local optimization feature, the OPLS3e force field, and adopting the Variable Dielectric Surface Generalized Born (VSGB) continuum solvation model. Prime MM-GBSA calculates the energy of each individual protein and the complex as a unit. The total predicted binding free energy for the complexes was then calculated using:

$$\Delta G_{binding} = G_{complex} - (G_{Sprotein} + G_{hACE2}) \quad (1)$$

$$\Delta G_{binding} = \Delta E_{MM} + \Delta G_{GB} + \Delta G_{SA} \quad (2)$$

$$\Delta E_{MM} = \Delta E_{Electrostatic} + \Delta E_{internal} + \Delta E_{VdW} \quad (3)$$

These calculations involve various energy terms, including molecular mechanics (MM), electrostatic, van der Waals, and solvation energies. $\Delta E_{MM}$ represents the total gas phase energy in OPLS3e. $\Delta E_{internal}$ includes bond, angle, and dihedral terms. These calculations incorporated contributions from gas phase energy, electrostatic interactions, van der Waals forces, and solvation effects via the generalized Born (GB) method and a nonpolar contribution. We conducted large-scale low-mode conformational sampling using MacroModel[24,25,75]. We employed the OPLS4[76] force field, a convergence threshold of 1.0, and an energy window for saving structures of 21.0 kJ/mol, along with a maximum atom deviation cutoff of 0.5 Å. We used the NetNGlyc 1.0 server[77] to calculate the glycosylation propensity. The structure files and MM/GBSA energy components, such as Coulomb, vDW, solvent, complex, are included in the CSV files on our Open Science Framework Page (https://osf.io/82n73/) and on Zenodo (https://zenodo.org/records/11404190).

## Statistical analyses

We used the "sf"[78] and "exactextractr"[79] packages in program R (4.1.1) to calculate the mean of the urbanization variables within each of our trap buffers. We calculated prevalence and 95% confidence intervals using the "PropCIs" package[80] with the Agresti-Coull method due to small sample sizes. We used the "glmmTMB" package[81] in program R to run univariate generalized linear mixed models to assess whether seroprevalence rates were impacted by the following covariates: urbanization (imperviousness), human presence, and resident population and included species as a random effect. All P-values presented represent a 2-tailed test.

## Reporting summary

Further information on research design is available in the Nature Portfolio Reporting Summary linked to this article.

## Data availability

All study data are included in the article and supporting information. Free Energy calculations can be accessed at https://doi.org/10.5281/zenodo.11404189. Sequence data from SARS-CoV-2 viruses collected from wildlife and sequenced in this study are available in the NCBI SARS-CoV-2 sequence repository. Accession numbers can be found in Supplemental Tables 2 and 3 and as follows OR866905, OR878666, OR866349, OR866382, OR866443, OR878668, OR866910, OR878667, OR866437, OR871756, OR872533, OR871072, OR871750, OR871751.

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

## Acknowledgements

We thank the following for land access: J. Eustis and New River Land Trust; B. Brodie, J. Jones and Mountain Lake Biological Research Station; R. Powers, K. Slusher, A. McGee and Roanoke Parks and Recreation; V. Malzone, G. Gorecki, S. Sweeney, J. Elliott, and Virginia State Parks; G. O'Malley, A. Lewis, H. Wander, K. Malewicz, V. Corbin, R. Cohen, S. Whitehead, N. Laggan, A. Grimaudo, and G. Blanvillain. We thank the following for help with study design and collection: M. Fisher, J. Ivan, K. Bentler, S. Yamada, C. Cereghino, K. Pierce and the Wildlife Center of Virginia, J. Riley and the Blue Ridge Wildlife Center, and H. Olsen-Hodges and the Southwest Virginia Wildlife Center. Additionally, we thank G. Eastwood, J. Tracey, A. Gomez, A. M. Kilpatrick, and NEON Biorepository for sharing samples with us. We thank J. Lemkul for helpful advice on the manuscript and M. Hellier, J. White, and M. Dwyer for assistance with fieldwork. This project was supported by funds from NSF DEB-1911853 to K.E.L. and J.R.H., the Virginia Tech FBRI to C.V.F., One Health award from VCOM to J.W, and the American Rescue Plan Act through USDA APHIS to K.E.L, J.W., C.V.F., and J.R.H. The findings and conclusions in this publication are those of the authors and should not be construed to represent any official USDA or U.S. government determination or policy. We gratefully acknowledge all data contributors, including authors and their originating laboratories responsible for obtaining the specimens, and the submitting laboratories for generating the genetic sequence and metadata and sharing via the GISAID Initiative, on which this research is based. All genome sequences and associated metadata in this dataset are published in GISAID's EpiCoV database. To view the contributors of each individual sequence with details such as accession number, virus name, collection date, originating lab and submitting lab and the list of authors, visit 10.55876/gis8.231109hc.

## Author contributions

A.R.G., K.E.L., and J.R.H. designed field research; A.R.G., C.D.K., M.J.K, K.E.L, and J.R.H. performed field research; A.R.G., J.M., P.R, K.E.L., J.W, K.L.B., C.R, A.C., R.B., C.V.F., and J.R.H. contributed to field data analysis; C.R., R.B., A.K.S., K.M.K., A.C., M.U., A.M.B., K.L.B., and C.V.F contributed to molecular and genomic analysis; K.E.L., J.W., C.V.F., and J.R.H. provided project administration; Members of the Molecular Diagnostics Laboratory provided technical support; and all authors aided in interpreting results and to writing and reviewing the paper.

## Competing interests

The authors declare no competing interests.
