## [Peer Review File · Nature Communications]

Widespread exposure to SARS-CoV-2 in wildlife communitiesREVIEWER COMMENTS

Reviewer #1 (Remarks to the Author):

Goldberg et al. evaluate the viral and serological evidence for SARS-CoV-2 infection in a total of 23 wild animal species that were caught in Virginia in 2022 and 2023. Virus was detected in oropharyngeal or nasopharyngeal swabs by RT-qPCR against 3 viral targets, providing convincing data of viral replication in six of 23 species examined, often with multiple positive individuals within each species. Further suggestion (although considered not conclusive) of wider infection among the species sampled was suggested by RT-qPCR positive results against just one of the PCR target sequences, with individuals from all six confirmed species and seven additional species showing a positive outcome. Sera was also collected from 6 species and neutralization titers against virus confirmed in individuals from 5 of these species, which suggests viral replication in two animal species that were not identified in the 2-target confirmed results of the RT-qPCR results. Analysis of the relationships between the presence of humans, termed imperviousness, and human presence, showed a positive correspondence between human presence and detection of virus in wild animals. And, further to this was sequence confirmation from about one half of the PCR positive samples confirmed multiple introductions of virus strains that were currently circulating in humans, suggesting incidence in these wild species was primarily related to cross-species transmission from humans to animals.

The data are very interesting and provide confirmation of the spread of SARS-CoV-2 into several wild animal species that were not previously known to be susceptible. This has important implications for identifying potential new reservoirs of the virus and for the potential contribute to viral evolution through adaptation in new species with potential for spread in the wild and reverse zoonosis. Specific comments regarding the manuscript are listed below.

Major comments:

1. Seroprevalance is an important criterion to use to establish prior infection in animals, and it is probably more likely that prior infection could be detected by the presence of virus-specific antibodies than by actual detection of ongoing or very recent infection through the detection of viral RNA using RT-qPCR. For this reason is it is somewhat surprising that a neutralization assay was used to detect virus specific antibodies as it would require higher titers of antibodies specific to spike for detection than if an ELISA based assay was used. Of course species specific secondary antibodies for Indirect ELISA are unlikely to be available for most animal species studied but sensitive methods like Double Antigen Sandwich ELISA, which are commercially available, have been used successfully in similar sorts of studies on seroprevalence, and can be used to target not only spike specific antibodies but also antibodies to nucleoprotein, which are typically very abundant, to provide a more sensitive indicator of prior infection with SARS-CoV-2. Some discussion/rationale of the selection of method used to identify virus-specific should be provided.
2. Why was no virus culture of RT-qPCR positive samples attempted? It would be very interesting to know whether there was detectable shedding of live virus since RNA levels tend to be much higher than titers of live virus and do not necessarily indicate significant live titers, at least in experimental animals. Shedding of live virus would suggest whether animals species were capable of animal to animal or animal to human spread and contribute to further understanding the risk that wild species infections pose.

Minor comments

1. References 12 and 19 are the same.
2. Figure 1d and Methods lines 491-6. It is not clear what is meant by the levels of neutralization used to define strong or weak positive vs negative. Is this based on reduction in plaque counts with complete absence of 90% (strong +) or 60% of plaques (weak +) or is this defined by reduction in plaque sizes. This should be defined more clearly.
3. In figure 1c, the meaning of the "Average Dilution" legend is unclear. Do the numbers 25, 50 and 75 mean 1:25, 1:50 and 1:75 dilutions to get at least 60% neutralization?
4. Line 92-3 (Figure 1 legend) – black is mentioned twice.
5. Figure 3. It is unclear why two different designations (Pango like XBB and NextStrain Clade like 23A) are being used together to refer to the same (at least in some instances) viruses. It would be good to at least provide some indication of how they overlap (i.e. XBB is 22F, XBB1.5 is 23A) to help with the relationship between them.

6. Line 482. 1300PFU mixed with serum dilutions is quite a large amount of virus when attempting to design an assay that is just sensitive enough to detect neutralization. For example mixing 50-100PFU with serum dilutions and plating all of it could be more sensitive to the amount of neutralizing activity. If the assay is based on a standardized or referenced protocol this should be stated.

7. Line 510-12, it is stated "Brush Mountain recreational area was closed to the public from March through August 2022 and thus, we assumed approximately 10 people used the area per month". It is unclear how this estimate is derived. The estimates for other sites span the period of March to August and since Brush Mountain is closed in the same time frame, it appears 0 people would have used the area between March and August. This should be explained better.

8. Line 520-21. How is the multiplier of 30 times the population derived in this estimation of number of people that used the area? Is this to suggest that the flow of people in and out the area is 30 times the number that live there continuously. The basis for this assumption is not clear and should be explained.

Reviewer #2 (Remarks to the Author):

The study sought to investigate SARS-CoV-2 spillovers in to wildlife species. The investigators used molecular detection and serological techniques to answer the questions. The detection of SARS-CoV-2 in free ranging host species other than white-tailed deer may be important to future pandemic prevention efforts. There are several studies looking a SARS-CoV-2 in many of these species, which did not result in viral detection (for which the reasons could be many). Overall, there are some substantial limitations.

The authors state that study "greatly expanded the known host range of SARS-CoV-2," but this statement must be questioned. As the authors also note, there is a very real possibility that some these SARS-CoV-2 detections do not reflect active infections. All of the detections occur with rather high Ct values on their PCR testing, and only 1/3 of their samples produced sequence coverage >80% of the SARS-CoV-2 genome. Undoubtedly there was SARS-CoV-2 RNA in 23 samples, but the claim of infection is weak. The claim would be much stronger with the recovery of viable virus from these host species, but it appears as though the investigators made that impossible with their selection of viral transport media (major study design flaw). This reviewer questions why the team would go to the effort to sample all of these animals, but not use VTM suitable for viral recovery. Several places in the manuscript the authors use the terms "isolate" and "isolated" when not only was virus isolation not attempted, it would have been impossible due to their VTM.

The apparent high identity to SARS-CoV-2 circulating in humans concurrent to sample collection indicates little to no animal-to-animal transmission in these species, which means these may be dead end hosts (limits the significance of these findings).

Serology: Neutralization was performed using a delta lineage virus. Given the expected life span of these species and the sample collection window, why not use omicron lineage?

The materials and methods mention testing 13,221 human clinical samples, but there seem to be no details about the origins of those samples. There is also no mention of IRB approval or exemption for use of these samples in research.

I am not qualified to evaluate the phylogenetic methods.

Reviewer #3 (Remarks to the Author):

The article by Goldberg et al. regarding the widespread of the SARS-CoV-2 in certain communities of wildlife. In order to quantify this study the authors have explore experimentally the collection of samples between 2022-2023 and quantify the presence of SARS-CoV-2 variants in different species as well as employed molecular modelling of the RBD:ACE2 receptor to validates their findings. The adaptation of variants in different wildlife communities certainly can pose a new pathway for the evolution of SARS-CoV-2 beside the currently known host (humans). In this regard the article seems to tackle an interesting topic. Certainly I find an exhaustive work in term o the RT-qPCR detections in years (2022-2023) as well as the comparison with the same species sampled both prior to SARS-CoV-2 arrival in 2020. A correlation seems to reported.

Regarding the in silico study starting in page 10, I am afraid the methodology employed and the analysis is not considering possible structural chagnes at the interface of the RBD:ACE2 and in S2 domain. Both responsible for the well-know stability of the RBD (see enhanced affinity doi.org/10.1038/s41467-021-27325-1 and additional mechanostability doi.org/10.1039/D0NR03969A). The authors have employed a common MD toolkit (i.e. Schrodinger) to mutate the S protein with energy minization and not conducted equilibration via MD simulation (i.e. NVT, NPT and long microsecond equilibration). I mention this because in previous studies has been reported the crucial effect of mutation in the dynamics of proteins. Authors should present a full equilibrated MD trajectory of the S protein with E471V and G798D mutations. After that, the MM/GBSA energy calculation which is a single point technique designed for small protein set has to be validated in several snapshots/frames and provide an error in the measure of energies. At this moment there is not statistical significance in the energy values reported.

In addition, the author should deposite the structures (several frames or trajectory) of the equilibrated models (e.g. zenodo repository) with mutations and a better description of the protocol of MM/GBSA calculation for the whole S protein in present of glycans and mutations.

The authors considered the glycosilation state of the S protein in the energy calculation and they do not provide a clear role.

I consider the need for a major revision with a better description of the dynamics of the S:ACE2 with those key mutations, Such study can provide a more in depth understanding.

Reviewer #4 (Remarks to the Author):

This study detected SARS-CoV-2 RNA and neutralizing antibodies in multiple species of wildlife in southwest Virginia. RNA was detected in deer mouse, Virginia opossum, raccoon, groundhog, Eastern cottontail rabbit, and Eastern red bat. They were able to sequence 12 SARS-CoV-2 genomes, allowing them to perform a more detailed phylogenetic analysis to examine the human-animal interface. All sequenced viruses were found to be different lineages of Omicron and appeared to be recent introductions from humans (possible mouse-to-mouse transmission). The detection of SARS-CoV-2 in so many new wildlife species is notable and should encourage more surveillance efforts targeting a wider range of species. The phylogenetic analysis, however, was not well presented and difficult to interpret.

1. The use of different PANGO and NextStrain nomenclatures, even in side-by-side panels (e.g., Figure 3), is problematic. PANGO lineages are used more commonly than NextStrain in the scientific literature, so better to go with that one.

2. The phylogenetic trees are unreadable. What do all the numbers mean? Please include strain names. Panel 3c is a nice summary, but where do those inferences come from? I'm confused by the question mark for the source of the BA.2.10.1 introduction into opossum. Why is it difficult to resolve the host species source for this introduction? What evidence is there that it is not from humans? Can you show the most closely related strains in a phylogenetic tree? Were the wildlife SARS-CoV2 sequences always most closely related to human SARS-CoV2 sequences from Virginia?

Or sometimes another state? And label them by host species? And please include strain names in these trees. How were these datasets selected? Are these the most closely related viruses from GISAID to each wildlife strain as identified in Usher? What is the outgroup? (It is also common practice to provide the raw tree files in a repository like GitHub.)

3. What methods were used to generate these trees? Maximum likelihood? What substitution models were used? Was a bootstrap analysis performed? NextStrain is a great exploratory tool for getting a snapshot look at data, but it's not designed to produce publication quality trees. IQTree can generate fast ML trees; or time-scale trees in BEAST can also provide estimates of when human-animal transferred occurred.

3. 12 SARS-CoV2 sequences were generated from wildlife for this study, but the supplementary table listing accession numbers only includes 9 viruses.

Isolate name Animal species Accession numbers

V0632866 Didelphis virginiana OR866905

V0654090 Didelphis virginiana OR878666

V0654124 Marmota monax OR866349

V0654196 Peromyscus maniculatus OR866382

V0654535 Peromyscus maniculatus OR866443

V0654612 Peromyscus maniculatus OR878668

V0655023 Peromyscus maniculatus OR866910

V0654185 Procyon lotor OR878667

V0654414 Sylvilagus floridanus OR866437

4. For the serology, could you perform a sensitivity analysis for different thresholds? How many samples would be considered positive if the threshold was lowered to 50%?

5. It would be nice to have a summary table with a row for each wildlife species tested and columns for (a) total number of animals tested and whether any animals were positive for SARS-CoV-2 by (b) serology; (c) PCR; (d) WGS; (e) partial sequence.

6. Is it possible to tell from the serology whether the animal was infected with omicron versus an earlier variant? If animals were sampled at different time points, could you infer anything about the duration of antibody responses?

7. Was the rate of SARS-CoV-2 detection higher at wildlife rehabilitation centers? How soon after capture was testing performed?

8. References 7 and 45 are the same.

REVIEWER COMMENTS:

Reviewer #1 (Remarks to the Author):

Goldberg et al. evaluate the viral and serological evidence for SARS-CoV-2 infection in a total of 23 wild animal species that were caught in Virginia in 2022 and 2023. Virus was detected in oropharyngeal or nasopharyngeal swabs by RT-qPCR against 3 viral targets, providing convincing data of viral replication in six of 23 species examined, often with multiple positive individuals within each species. Further suggestion (although considered not conclusive) of wider infection among the species sampled was suggested by RT-qPCR positive results against just one of the PCR target sequences, with individuals from all six confirmed species and seven additional species showing a positive outcome. Sera was also collected from 6 species and neutralization titers against virus confirmed in individuals from 5 of these species, which suggests viral replication in two animal species that were not identified in the 2-target confirmed results of the RT-qPCR results. Analysis of the relationships between the presence of humans, termed imperviousness, and human presence, showed a positive correspondence between human presence and detection of virus in wild animals. And, further to this was sequence confirmation from about one half of the PCR positive samples confirmed multiple introductions of virus strains that were currently circulating in humans, suggesting incidence in these wild species was primarily related to cross-species transmission from humans to animals.

The data are very interesting and provide confirmation of the spread of SARS-CoV-2 into several wild animal species that were not previously known to be susceptible. This has important implications for identifying potential new reservoirs of the virus and for the potential contribute to viral evolution through adaptation in new species with potential for spread in the wild and reverse zoonosis. Specific comments regarding the manuscript are listed below.

Major comments:

1. Seroprevalence is an important criterion to use to establish prior infection in animals, and it is probably more likely that prior infection could be detected by the presence of virus-specific antibodies than by actual detection of ongoing or very recent infection through the detection of viral RNA using RT-qPCR. For this reason, is it is somewhat surprising that a neutralization assay was used to detect virus specific antibodies as it would require higher titers of antibodies specific to spike for detection than if an ELISA based assay was used. Of course species specific secondary antibodies for Indirect ELISA are unlikely to be available for most animal species studied but sensitive methods like Double Antigen Sandwich ELISA, which are commercially available, have been used successfully in similar sorts of studies on seroprevalence, and can be used to target not only spike specific antibodies but also antibodies to nucleoprotein, which are typically very abundant, to provide a more sensitive indicator of prior infection with SARS-CoV-2. Some discussion/rationale of the selection of method used to identify virus-specific should be provided.

Response: We appreciate this suggestion. While neutralization assays are possibly less sensitive than an ELISA-based approach, they are also much more specific since non-neutralizing antibodies are highly cross-reactive between coronaviruses ([https://www.thelancet.com/journals/ebiom/article/PIIS2352-3964\(21\)00494-1/fulltext](https://www.thelancet.com/journals/ebiom/article/PIIS2352-3964(21)00494-1/fulltext)).

Surprisingly, false positives have also been observed in binding-based assays in patients with other infections (<https://www.nature.com/articles/s41598-022-26709-7>) or even auto-immune diseases (<https://www.frontiersin.org/articles/10.3389/fimmu.2021.666114/full>). Thus, while there are potential benefits to using ELISA, neutralization tests still provide the most specific means of assessing prior exposure to SARS-CoV-2. We have added this rationale into the discussion.

2. Why was no virus culture of RT-qPCR positive samples attempted? It would be very interesting to know whether there was detectable shedding of live virus since RNA levels tend to be much higher than titers of live virus and do not necessarily indicate significant live titers, at least in experimental animals. Shedding of live virus would suggest whether animal species were capable of animal to animal or animal to human spread and contribute to further understanding the risk that wild species infections pose.

Response: We did consider collecting additional samples to try and isolate live virus from any animals that tested positive. However, we ended up using a viral transport media that rendered any virus non-viable because of the added health and safety challenges for collecting and transporting potentially live virus and the subsequent time and money investment in processing of these samples. Given that samples were also collected from field locations, there were also additional considerations for preservation, transport, and maintenance of samples from some sites to the lab. Since we initially expected quite low SARS-CoV-2 prevalence, we selected an approach that optimized along these axes (e.g. health, safety, logistics, and costs). However, we agree with the reviewer that this would be helpful in the future and are working on a way we can collect a subset of samples that we could collect live virus from going forward.

Minor comments

1. References 12 and 19 are the same.

Response: Thank you very much for catching this. We have corrected the mistake.

2. Figure 1d and Methods lines 491-6. It is not clear what is meant by the levels of neutralization used to define strong or weak positive vs negative. Is this based on reduction in plaque counts with complete absence of 90% (strong +) or 60% of plaques (weak +) or is this defined by reduction in plaque sizes. This should be defined more clearly.

Response: The level of neutralization is based on the reduction of plaque counts and not the plaque size. We have updated the methods to clarify our analysis.

3. In figure 1c, the meaning of the “Average Dilution” legend is unclear. Do the numbers 25, 50 and 75 mean 1:25, 1:50 and 1:75 dilutions to get at least 60% neutralization?

Response: We apologize for any confusion. This is referring to the percent neutralization of a sample, which is indicated by the size of the circle. We have revised the figure text to more accurately reflect this and have added clarification to the figure legend.

4. Line 92-3 (Figure 1 legend) – black is mentioned twice.

Response: Thank you very much for catching this. We have removed the duplicate word.

5. Figure 3. It is unclear why two different designations (Pango like XBB and NextStrain Clade like 23A) are being used together to refer to the same (at least in some instances) viruses. It would be good to at least provide some indication of how they overlap (i.e. XBB is 22F, XBB1.5 is 23A) to help with the relationship between them.

Response: Thank you very much for pointing out that area of confusion. We agree and have removed NextStrain clades and now just use Pango lineages to make it easier to relate each of the panels within Figure 3 to one another and to address another reviewer's comments.

6. Line 482. 1300PFU mixed with serum dilutions is quite a large amount of virus when attempting to design an assay that is just sensitive enough to detect neutralization. For example, mixing 50-100PFU with serum dilutions and plating all of it could be more sensitive to the amount of neutralizing activity. If the assay is based on a standardized or referenced protocol this should be stated.

Response: We generated a solution of SARS-CoV-2 at a concentration of 1300 PFU/mL, which resulted in us adding ~40 PFUs per well, just below the range you suggested. We have added additional text to clarify this in the methods. This amount is consistent with various other reports (<https://www.nature.com/articles/s41467-020-17892-0> and <https://journals.plos.org/plosbiology/article?id=10.1371/journal.pbio.3001284>)

7. Line 510-12, it is stated “Brush Mountain recreational area was closed to the public from March through August 2022 and thus, we assumed approximately 10 people used the area per month”. It is unclear how this estimate is derived. The estimates for other sites span the period of March to August and since Brush Mountain is closed in the same time frame, it appears 0 people would have used the area between March and August. This should be explained better.

Response: We have added some clarifying text to the methods to better explain how we came up with this estimate. All the estimates were on the same time scale: # of persons/month averaged over March – August 2022. We have added more detail about how we derived the estimate of 10 people/month for Brush Mountain. There was one person utilizing the area occasionally to remove trees for future trail work but they were only on the property a few days a month (personal communication with the property manager). We then assumed a few other visits by locals who may occasionally walk through the area (observed one person while working at the site) for recreation despite the fact that it was technically closed. Hence, we estimated about 10 visits per month on average between March and August 2022.

8. Line 520-21. How is the multiplier of 30 times the population derived in this estimation of number of people that used the area? Is this to suggest that the flow of people in and out the area is 30 times the number that live there continuously. The basis for this assumption is not clear and should be explained.

Response: We apologize for the confusion. We are estimating the number of people using an area per month. As a result, we are assuming that the population is relatively constant each day, so we multiplied it by 30 to represent 30 days of people occupying an area. We agree there likely are some days where more people are in town (weekend) or fewer (weekdays), but we are working under the assumption that the average monthly usage is about 30 times the population. We have added additional text to the methods to make this more clear.

Reviewer #2 (Remarks to the Author):

The study sought to investigate SARS-CoV-2 spillovers into wildlife species. The investigators used molecular detection and serological techniques to answer the questions. The detection of SARS-CoV-2 in free ranging host species other than white-tailed deer may be important to future pandemic prevention efforts. There are several studies looking a SARS-CoV-2 in many of these species, which did not result in viral detection (for which the reasons could be many). Overall, there are some substantial limitations.

1. The authors state that study "greatly expanded the known host range of SARS-CoV-2," but this statement must be questioned. As the authors also note, there is a very real possibility that some these SARS-CoV-2 detections do not reflect active infections. All of the detections occur with rather high Ct values on their PCR testing, and only 1/3 of their samples produced sequence coverage >80% of the SARS-CoV-2 genome. Undoubtedly there was SARS-CoV-2 RNA in 23 samples, but the claim of infection is weak. The claim would be much stronger with the recovery of viable virus from these host species, but it appears as though the investigators made that impossible with their selection of viral transport media (major study design flaw). This reviewer questions why the team would go to the effort to sample all of these animals, but not use VTM suitable for viral recovery. Several places in the manuscript the authors use the terms "isolate" and "isolated" when not only was virus isolation not attempted, it would have been impossible due to their VTM.

Response: We apologize for the misuse of “isolate/isolated” in the manuscript. We have removed the 4 instances where this was not applicable and have replaced them with more appropriate terminology (collected).

We thank the reviewer for the comments concerning live virus collection and agree that there are limitations to the interpretation of qPCR positive detections, which we have included throughout the manuscript. However, given the specificity of our qPCR, we can definitively conclude that SARS-CoV-2 is being detected in a larger range of hosts than previously reported, with important implications for the ecology of this disease and

ongoing global surveillance efforts. While live virus was not isolated, we want to highlight that this would still be steps away from demonstrating animal-to-animal transmission, which would require experimental infections and subsequent transmission experiments, which was outside of the scope of our current project. We did discuss and consider collecting additional samples to try and isolate live virus from any animals that tested positive (see response #2 to reviewer 1). Ultimately, we decided not to go this route because of the added health and safety restrictions for collecting and transporting potentially live viruses from wild animals, the subsequent investment in processing of these samples. In addition, we would face limitations in interpreting the genomic data from virus that was isolated from animals and propagated in culture, as we would be unable to determine whether mutations emerged as result of that procedure. We think our findings still represent a significant result, given that many species are being exposed to SARS-CoV-2 in the wild. It demonstrates that human-to-animal transmission routes exist for a wide array of species, and there is high probability these species will eventually be exposed to a lineage that is capable of infecting them. This could result in viral shedding and subsequent transmission, if it has not already (see response below). Ultimately, we agree with the reviewer and are working on a way we can collect even a subset of samples so we can isolate live virus going forward.

2. The apparent high identity to SARS-CoV-2 circulating in humans concurrent to sample collection indicates little to no animal-to-animal transmission in these species, which means these may be dead end hosts (limits the significance of these findings).

Response: We agree that our findings indicate a large number of human-to-animal transmission events but have no evidence to suggest these species are dead-end hosts, particularly because several of them have been shown to be capable of transmission relevant shedding rates in the lab (e.g. deer mice; Table S9). In addition, while some hosts might not be less important for transmission, wildlife can still suffer disease impacts, as has been observed in mink, and thus documenting SARS-CoV-2 exposure in wild mammals has conservation relevance. Lastly, as mentioned above, if exposure continues at the rate we observed, there is a high probability that some of these species will eventually be exposed to a lineage that can cause animal-to-animal transmission. We also did find two instances where the phylogeny is unresolved (an opossum and deer mouse), which could indicate animal-to-animal transmission. We tried to be abundantly conservative in our interpretation of these data but given the information we have and experimental work on these species, animal-to-animal transmission may be the most parsimonious explanation.

3. Serology: Neutralization was performed using a delta lineage virus. Given the expected life span of these species and the sample collection window, why not use omicron lineage?

Response: We collected our serum samples in 2022, which was just after the end of the Delta wave so we assumed many of the neutralizing antibodies in these captured animals would likely reflect prior exposure to the Delta variant. Given the life expectancy of the species we were sampling, they would have extended well into the period that the Delta variant was circulating. We have added the table below to the supplementary materials (Supplementary Table 12), and additional justification into the methods.

Species	Average lifespan in the wild (years); max is in parenthesis
Procyon lotor	5 ^a (20 ^b)
Didelphis virginiana	1.5-2 ^a
Mephitis mephitis	<1 ^a (6 ^b)
Sciurus carolinensis	(12.5 ^b)
Peromyscus maniculatus	<1 ^a
Peromyscus leucopus	1 ^a
Blarina brevicauda	(2.5 ^b)
Marmota monax	4-6 ^a
Lasiurus borealis	
Sylvilagus floridanus	<3 ^a (5 ^b)
Tamias striatus	<2 ^a (3 ^b)
Vulpes vulpes	3 ^a (7 ^b)

^aMyers, P., R. Espinosa, C. S. Parr, T. Jones, G. S. Hammond, and T. A. Dewey. 2024. The Animal Diversity Web (online). Accessed at <https://animaldiversity.org>

^bCarey, J. R., & Judge, D. S. 2002. Longevity records: Life spans of mammals, birds, amphibians, reptiles, and fish. Monographs on population aging, (8).

4. The materials and methods mention testing 13,221 human clinical samples, but there seem to be no details about the origins of those samples. There is also no mention of IRB approval or exemption for use of these samples in research.

Response: We have added additional details into the methods about the human clinical samples and the IRB that this sample collection was conducted under. Samples were collected by the Virginia Department of Health and submitted to the Molecular Diagnostics Laboratory as part of a genomic surveillance agreement.

Reviewer #3 (Remarks to the Author):

The article by Goldberg et al. regarding the widespread of the SARS-CoV-2 in certain communities of wildlife. In order to quantify this study, the authors have explored experimentally the collection of samples between 2022-2023 and quantify the presence of SARS-CoV-2 variants in different species as well as employed molecular modelling of the RBD:ACE2 receptor to validates their findings. The adaptation of variants in different wildlife communities certainly can pose a new pathway for the evolution of SARS-CoV-2 beside the currently known host (humans). In this regard the article seems to tackle an interesting topic. Certainly I find an exhaustive work in term of the RT-qPCR detections in years (2022-2023) as well as the comparison with the same species sampled both prior to SARS-CoV-2 arrival in 2020. A correlation seems to reported.

The authors appreciate the reviewer's comments and recognition of the work presented in this article by our group.

1. Regarding the in-silico study starting in page 10, I am afraid the methodology employed, and the analysis is not considering possible structural changes at the interface of the RBD:ACE2 and in S2 domain. Both responsible for the well-known stability of the RBD (see enhanced affinity doi.org/10.1038/s41467-021-27325-1 and additional mechanostability doi.org/10.1039/D0NR03969A).

Response: The authors acknowledge the importance of considering possible structural changes and dynamics at the interface of RBD:ACE2 and in the S2 domain, as they are relevant to the stability of RBD. In response to the reviewer's suggestion, we have now included contextualization and discussion of the relevant literature regarding the dynamics of the RBD in relation to ACE2 binding.

The new text now reads as follow:

“While this approach limits the investigation of these mutations and their role on dynamic conformational states of S protein, it lays the foundation for understanding the mechanistic impact and connection of structure to experimental observables. Future work will involve molecular dynamics (MD) simulations to assess the impact of these mutations on structural stability, membrane interaction, and conformational states. This research will contribute to our understanding of evolutionarily pressures and cross-species transmission.”

2. The authors have employed a common MD toolkit (i.e. Schrodinger) to mutate the S protein with energy minimization and not conducted equilibration via MD simulation (i.e. NVT, NPT and long microsecond equilibration). I mention this because in previous studies has been reported the crucial effect of mutation in the dynamics of proteins. Authors should present a full equilibrated MD trajectory of the S protein with E471V and G798D mutations.

Response: We appreciate your suggestion regarding the equilibration process and thank the reviewer for thoughtful comments. While we utilize the Schrodinger toolkit for mutation and energy minimization, we acknowledge the importance of presenting a fully equilibrated MD trajectory for a comprehensive understanding of the dynamics, especially in light of previous studies emphasizing the effect of mutations. However, performing MD simulations for over 1 microsecond on multiple mutations within the complete Spike:ACE2 complex to fully describe the dynamic events is not feasible within a reasonable time frame (it would have taken roughly 6 to 10 months using all the computing resources at Virginia Tech based on our calculations) due to the scale of the entire system (1 million+ atoms) and the scope of this study. Moreover, it is not suitable for this work to only simulate the clipped RBD + S protein, as including the S2 region is essential for clarity and comprehensiveness in a thorough MD investigation. Further MD simulation studies would be better suited as standalone papers to address the combined impact of mutation and provide a comprehensive discussion of the dynamic state of Spike bound to ACE2. In this work, our goal was to offer context and initial rationalization of the location and interaction impact between the Spike RBD and ACE2, focusing on the E471V and local structural impact of the G798D mutations. Indeed, for E471V, the local point-based mutations had an impact on the interaction energy, highlighting a potential enthalpic contribution of this mutation on the RBD interface interacting with ACE2. We have

incorporated this additional context into the manuscript and will carefully consider your recommendation in a follow-up work.

The text now reads as follow:

“We conducted in silico studies to probe the local impact of the E⁴⁷¹V and G⁷⁹⁸D mutations on the S protein. We employed molecular modeling and molecular mechanics/generalized born surface area (MM/GBSA) free energy calculations to predict the free energy of binding interaction (ΔG_{bind}). It is important to highlight that our focus in this study is on probing the local impact of E⁴⁷¹V and G⁷⁹⁸D on the favorability of S protein-hACE2 interaction and the modifications in the S2 site as hypothesis for the experimentally observed traits found in SARS-CoV-2 wildlife reservoir.”

“The increased favorability of the E⁴⁷¹V mutation for hACE2 interaction aligns with the previously reported trend of residues in the RBD influencing stabilization and enhanced affinity (Koehler et al. 2021 Nat Comm). Any differences may be attributed to entropic effects, emphasizing the potential enthalpic contribution of this mutation to improve hACE2 interaction at the RBD interface.”

“Future work will involve molecular dynamics (MD) simulations to evaluate the impact of these mutations on structural stability, membrane interaction, and conformational states. This research will contribute to our understanding of evolutionarily pressures and cross-species transmission.”

3. After that, the MM/GBSA energy calculation which is a single point technique designed for small protein set has to be validated in several snapshots/frames and provide an error in the measure of energies. At this moment there is not statistical significance in the energy values reported.

Response: To generate multiple conformations for statistical analysis, we conducted large-scale low-mode conformational sampling on the RBD (residues 329-531) of both BA.2 and E⁴⁷¹V Spike in complex with hACE2 (Watts et al. 2014 J Chem Info). To reduce computational expense, Spike was truncated to include only the RBD. The top five sampled conformations of E⁴⁷¹V-hACE2, based on energy, exhibited significantly more favorable binding interaction energy than BA.2-hACE2. This information has been included in the computational section for its statistical value.

The text now reads as follows:

“Large-scale low-mode conformational sampling was performed on the RBD of BA.2 and E⁴⁷¹V Spike (residues 329-531) with hACE2 to probe ΔG_{bind} across multiple potential conformations (Watts et al. 2014 J Chem Info, Kolossvary 1996 J Amer Chem Soc). Consistent with the results observed in full-length Spike-hACE2 interactions, the mean ΔG_{bind} for the top five most favorable E⁴⁷¹V_{RBD} conformations (-151.7 ± 19.2 kcal/mol) was more favorable than the mean for BA.2_{RBD} conformations (125.4 ± 4.1 kcal/mol, $p \leq 0.05$). The sampled conformations of E⁴⁷¹V_{RBD} exhibited a larger range of ΔG_{bind} values relative to

BA.2_{RBD}, reflecting not only increased interaction flexibility but also increased flexibility in the RBD due to the E⁴⁷¹V mutation.”

The methods section has been updated to reflect these changes:

“We conducted large-scale low-mode conformational sampling using MacroModel. The OPLS4 force field, a convergence threshold of 1.0, and an energy window for saving structures of 21.0 kJ/mol, along with a maximum atom deviation cutoff of 0.5 Å, were employed.”

4. In addition, the author should deposit the structures (several frames or trajectory) of the equilibrated models (e.g. zenodo repository) with mutations and a better description of the protocol of MM/GBSA calculation for the whole S protein in present of glycans and mutations.

Response: At the reviewer’s request, we have deposited the structures and data files, along with detailed information on the MM/GBSA calculation protocol for the entire S protein in the presence of glycans and mutations. These files are accessible on our Open Science Framework repository at <https://osf.io/82n73/>. Additionally, we have expanded the methods section to provide further clarity on the conducted MM/GBSA calculation procedure.

The text now reads as follows:

“In brief, the S protein and hACE2 complex underwent a second round of minimization using the local optimization feature, the OPLS3e force field, and adopting the Variable Dielectric Surface Generalized Born (VSGB) continuum solvation model. Prime MM-GBSA calculates the energy of each individual protein and the complex as a unit. The total predicted binding free energy for the complexes was then calculated using:

$$\Delta G_{\text{binding}} = G_{\text{complex}} - (G_{\text{S protein}} + G_{\text{hACE2}})$$

$$\Delta G_{\text{binding}} = \Delta E_{\text{MM}} + \Delta G_{\text{GB}} + \Delta G_{\text{SA}}$$

$$\Delta E_{\text{MM}} = \Delta E_{\text{Electrostatic}} + \Delta E_{\text{internal}} + \Delta E_{\text{vdW}}$$

These calculations involve various energy terms, including molecular mechanics (MM), electrostatic, van der Waals, and solvation energies. ΔE_{MM} represents the total gas phase energy in OPLS3e. $\Delta E_{\text{internal}}$ includes bond, angle, and dihedral terms. These calculations incorporated contributions from gas phase energy, electrostatic interactions, van der Waals forces, and solvation effects via the generalized Born (GB) method and a nonpolar contribution.”

5. The authors considered the glycosylation state of the S protein in the energy calculation and they do not provide a clear role.

Response: We appreciate the reviewer’s attention to the glycosylation state of the S protein in our energy calculation. We have now extended our explanation regarding the role of glycosylation, particularly at the S2 site. The inclusion of glycosylation sites in our structural investigation aligns with previous work (<https://www.ncbi.nlm.nih.gov/pmc/articles/PMC8260498/>) and provides an important context for understanding structure-based modifications. We trust this additional clarification enhances the comprehensiveness of our study.

The text now reads as follows:

“When glycosylated, the N⁸⁰¹ glycosylation site significantly influences the reduction in viral entry, with very low mutation rates observed (e.g., within 2-3 residues from position 801). Interestingly, D⁷⁹⁸ creates a small, charged pocket on this solvent-accessible S2 loop, decreasing the probability of glycosylation at N⁸⁰¹ (0.48) compared to BA.2 [0.61]²⁹. It is hypothesized that G⁷⁹⁸D could impact structural stability and membrane interaction³⁰ (Fig. 4 and Fig. S10) and have a negative impact on Spike maturation and infectivity. While this approach limits the investigation of these mutations and their role in the dynamic conformational states of the S protein, it provides a foundation for understanding the mechanistic impact and connection of structure to experimental observables.”

“Glycosylation propensity was calculated using the NetNGlyc 1.0 server⁶⁸. The structure files and MM/GBSA energy components, such as Coulomb, vDW, solvent, complex, are included in the CSV files on our Open Science Framework Page (<https://osf.io/82n73/>).”

6. I consider the need for a major revision with a better description of the dynamics of the S:ACE2 with those key mutations, such study can provide a more in depth understanding.

Response: We thank the reviewer for their thoughtful and thorough review of the manuscript and their reconsideration for major revisions.

Reviewer #4 (Remarks to the Author):

This study detected SARS-CoV-2 RNA and neutralizing antibodies in multiple species of wildlife in southwest Virginia. RNA was detected in deer mouse, Virginia opossum, raccoon, groundhog, Eastern cottontail rabbit, and Eastern red bat. They were able to sequence 12 SARS-CoV-2 genomes, allowing them to perform a more detailed phylogenetic analysis to examine the human-animal interface. All sequenced viruses were found to be different lineages of Omicron and appeared to be recent introductions from humans (possible mouse-to-mouse transmission). The detection of SARS-CoV-2 in so many new wildlife species is notable and should encourage more surveillance efforts targeting a wider range of species. The phylogenetic analysis, however, was not well presented and difficult to interpret.

1. The use of different PANGO and NextStrain nomenclatures, even in side-by-side panels (e.g., Figure 3), is problematic. PANGO lineages are used more commonly than NextStrain in the scientific literature, so better to go with that one.

Response: We appreciate this comment and we have changed Figure 3 to only present the Pango lineages and reduce any confusion.

2a. The phylogenetic trees are unreadable. What do all the numbers mean? Please include strain names.

Response: We apologize that the trees were too small to read. We have saved them in a taller format so they can be stretched to fit a page and hopefully make them easier to read.

We have also uploaded the main phylogenetic figure (Fig. 3) as a separate manuscript file, and included strain names, state of collection, and year when available for easier interpretation.

2b. Panel 3c is a nice summary, but where do those inferences come from? I'm confused by the question mark for the source of the BA.2.10.1 introduction into opossum. Why is it difficult to resolve the host species source for this introduction? What evidence is there that it is not from humans?

Response: For Figure 3c, the inferences about transmission were determined by phylogenetic analysis using local, related human host sequences and any of our sequences collected from wild animals as shown in Supplementary Figures 2-9 and in Supplementary Table 3. We also considered any unique mutations as those mutations which may have come from the originating host's virus.

For the two events where we included question marks, these were based on two different scenarios. For the opossum (BA.2.10.1), we reported a mutation in the sample collected that has not been identified in any human sample collected. This of course could be from a human that was never sampled, arisen in the opossum itself, or occurred in another animal. Given the uncertainty, we designed this as a question mark. For mice, we had two samples collected from mice (Fig. S8) that clustered together. Again, we cautiously interpreted this as either both mice were exposed to the same infected human, or the mouse was exposed to the other mouse. Given that deer mice have been shown experimentally to be capable of transmitting, we felt either of these scenarios could be likely. We have added additional text to the Fig. 3 legend to clarify how we determined whether the source was likely human or unknown.

2c. Can you show the most closely related strains in a phylogenetic tree?

Response: We have included all the most closely related strains in individual phylogenetic trees in the supplemental information (Fig. S2-S9). The two mice described above are included in the same tree. All of the SARS-CoV-2 sequences used for phylogenetic analysis were retrieved from human hosts except for our sequences collected from wild animals.

2d. Were the wildlife SARS-CoV2 sequences always most closely related to human SARS-CoV2 sequences from Virginia? Or sometimes another state?

Response: In some cases, yes. For example, the most closely matched human sample to the two deer mice mentioned above was from Vienna, VA. However, the opossum with the unique mutation has the closest match to a human sample collected from NY. We have now added the state abbreviation to all the branch tip names to aid in seeing the origin location. We also have the closest human sample and state of origin in Table S3.

2e. And label them by host species? And please include strain names in these trees.

Response: Phylogenetic trees have been modified to include strain names and host species are now indicated with color tips.

2f. How were these datasets selected? Are these the most closely related viruses from GISAID to each wildlife strain as identified in UshER? What is the outgroup? (It is also common practice to provide the raw tree files in a repository like GitHub.)

Response: All SARS-CoV-2 sequences used for phylogenetic analysis were retrieved from human hosts except for our sequences collected from wild animals. Selection of the datasets used for our phylogenetic trees are described in the Methods section lines 596-600 and 605-615. UshER was not used to select the sequences only to identify which PANGO lineages to use for which virus. All of our trees used the Reference Sequences for SARS-CoV-2 from NCBI (NC_045512.2). We appreciate the suggestion to submit the sequences used for phylogenetic analysis to GitHub for public availability. However, due to the use of GISAID sequences which are only available for registered users, we've listed the sequences for our analyses in Supplementary Table 11.

3. What methods were used to generate these trees? Maximum likelihood? What substitution models were used? Was a bootstrap analysis performed? NextStrain is a great exploratory tool for getting a snapshot look at data, but it's not designed to produce publication quality trees. IQTree can generate fast ML trees; or time-scale trees in BEAST can also provide estimates of when human-animal transferred occurred.

Response: Our phylogenetic trees are Maximum Likelihood (ML) trees generated using TreeLine (Methods line 601) in R. TreeLine selected the substitution model based on Akaike information criterion for each tree. Usually, GTR+G4 was selected as the substitution model. We have added "maximum likelihood" and the substitution model used to each phylogenetic tree's figure legend and to Supplementary Table 10. Boot strap analysis was not performed. We agree with the reviewer that NexStrain is a great tool for exploring data initially and so it wasn't used for tree generation in this manuscript. We used TreeLine to generate the tree and iTOL to view the trees (Methods line 601-602). We will explore the suggested programs (IQTree and BEAST) for our future phylogenetic needs to compare them to TreeLine.

3. 12 SARS-CoV2 sequences were generated from wildlife for this study, but the supplementary table listing accession numbers only includes 9 viruses.

Isolate name Animal species Accession numbers

V0632866 Didelphis virginiana OR866905

V0654090 Didelphis virginiana OR878666

V0654124 Marmota monax OR866349

V0654196 Peromyscus maniculatus OR866382

V0654535 Peromyscus maniculatus OR866443

V0654612 Peromyscus maniculatus OR878668

V0655023 Peromyscus maniculatus OR866910

V0654185 Procyon lotor OR878667
V0654414 Sylvilagus floridanus OR866437

Response: We apologize for any confusion the remaining 3 virus samples are in a separate Table (S4) since we were only able to get partial sequence. Accession numbers have been added to both Supplementary Tables 3 and 4.

4. For the serology, could you perform a sensitivity analysis for different thresholds? How many samples would be considered positive if the threshold was lowered to 50%?

Response: We re-ran the generalized linear mixed model using different cutoffs for percent neutralization (40, 50, 65, and 70), which is now available in Supplementary Table 6. We find statistical support for this relationship at all cut-offs except for the most conservative (70% neutralization). However, even at this higher cutoff, we continue to find a negative relationship between these two variables. Additionally, we have included the differences in percent positivity using different threshold cutoffs, and the sensitivity of our analysis examining the relationship between imperviousness and seroprevalence using these cutoffs in the table below.

% Neutralization cutoff chosen	Intercept			Imperviousness slope		
	β	SE	p	β	SE	p
40%	-0.054	0.372	0.884	0.053	0.024	0.026*
50%	-0.608	0.380	0.109	0.042	0.019	0.024*
60%	-1.655	0.477	0.001	0.038	0.018	0.032*
65%	-1.892	0.514	0.000	0.044	0.019	0.017*
70%	-1.885	0.561	0.001	0.005	0.179	0.858

5. It would be nice to have a summary table with a row for each wildlife species tested and columns for (a) total number of animals tested and whether any animals were positive for SARS-CoV-2 by (b) serology; (c) PCR; (d) WGS; (e) partial sequence.

Response: We agree and have added a table (Supplemental Table 7) that summarizes all of this information. Thank you for this suggestion.

6. Is it possible to tell from the serology whether the animal was infected with omicron versus an earlier variant? If animals were sampled at different time points, could you infer anything about the duration of antibody responses?

Response: This is a great question. Previous published work has suggested that certain variants can be distinguished based on the strength of neutralization, and this is something we are eager to understand in the future. We are hoping to obtain enough recaptured animals where we know what variant they were previously exposed to, so we can look at this in known individuals. Unfortunately, the serology data in this study was collected over a very brief period of time ~6 weeks, so we were not able to examine this question in this manuscript. We do hope to be able to address this question in future manuscripts when we will have completed regular sampling over a longer period of time with recaptured

individuals.

7. Was the rate of SARS-CoV-2 detection higher at wildlife rehabilitation centers? How soon after capture was testing performed?

Response: Thank you for that interesting question. The rehabilitation centers had 2.24% positivity across all species while we had 4.04% in our field collected animals. We will note that species like mice are not seen by wildlife rehabilitation centers which is the primary contributing factor to the differences. If we examine positivity rate by species with the greatest number of samples collected in both datasets: we find the following:

Species	Rehabilitation	Field data
Virginia opossum	2.83%	2.94%
Raccoon	3.39%	8.00%

It does suggest we have slightly higher positivity rates in the field compared to the rehabilitation clinics for raccoons. We would have hypothesized the opposite – animals are typically being brought to clinics because they are sick or injured. However, our field sampling is generally more intensive in a given area whereas submissions to rehab centers are a bit more diffuse. As you can see if Figure 3a, most of the positive samples were collected during fall 2023, and we had multiple positives at a site during a four-day trapping period. This means if there was a change in human infection rates or activity that results in more exposure in wildlife, we are more likely to get multiple positives at that time. We agree with the reviewer that these are important points for considering surveillance going forward and have added some text about this in the manuscript. For the timing of our sampling, animals were trapped overnight and then processed in the morning, for the rehabilitation centers, animals are brought to the center (typically within 24 hr. following encounters) and then processed upon arrival.

8. References 7 and 45 are the same.

Response: We have corrected this error.

REVIEWER COMMENTS

Reviewer #1 (Remarks to the Author):

I have assessed the responses of the authors to my original review of the manuscript. The authors had adequately addressed all comments and I have no further comments for consideration.

Reviewer #2 (Remarks to the Author):

My comments have been addressed to the extent possible given the study design.

Reviewer #3 (Remarks to the Author):

In the revised article I find some improvements but additional in silico work has to be done to improve the message of the MS.

Major comments;

Having snapshot of the complex system that have been only energy minimized are not sufficient to prove what they author suggest in the MS. I have suggested to perform common MD protocols with NVT and NPT equilibration runs, typical each of few nanoseconds. Regarding the dynamics of the whole S:ACE2 complex with mutations, it is important to consider time scale of microseconds. The authors described the difficulties in terms of computational cost. However, I believe there was a misunderstanding regarding the time scale, yes microsecond but one can consider several parallel runs each of 200 ns. In this way, one can have 5 independent replicas, better statistics and obtain some dynamics of the possible conformations induced by those mutations. As the author mentioned, it will take 6-10 months to runs 1 microsecond in Virgitina Tech HPC, well, in practice it may take less than a 1-2 month considering replica study. The suggested parallel study is common in MD.

I consider this computational study crucial, as in the section regarding free energy calculations. the authors now report values of free energy after perform large-scale low-mode conformational sampling which is another form of the simple NMA for large structures. The problem I can see is to have applied to a complex and not a single chain. For the case of the full S:ACE2 complex considering the mutations. The values are very contradictory as now they posses error bars and the absolute values are quite different from each other: E471V-RBD conformations (-151.7 ± 19.2 kcal/mol) and for BA.2-RBD conformations (125.4 ± 4.1 kcal/mol). Here the problem is reflected in the free energy difference, which is ~ 300 kcal/mol between both variants, this is totally off. I believe this is the results of having conformations from NMA which are not consistent with the true conformational landscape of the protein complex.

If the authors can provide some biophysical insight into the conformations via MD dynamics, it will simply enrich the whole message. I totally undestand the need to use MD, but it has to be carefully used.

Reviewer #4 (Remarks to the Author):

TreeLine is not a standard program for phylogenetic (only 41 citations compared to >6500 for IQTree2). I suspect because the trees are not so well resolved, with many polytomies like in Figure 3 and not able to run bootstraps. But this is not a phylogenetics paper and the tree in Figure 3 is not necessarily wrong. The addition of strain names is a great improvement.

Reviewer 3, Comment #1: *“In the revised article I find some improvements but additional in silico work has to be done to improve the message of the MS.”*

Authors’ response: The authors thank the reviewer for taking the time to review our manuscript and are pleased to learn that improvements were found.

Reviewer 3, Comment #2: *“Having snapshot of the complex system that have been only energy minimized are not sufficient to prove what they author suggest in the MS. I have suggested to perform common MD protocols with NVT and NPT equilibration runs, typical each of few nanoseconds. Regarding the dynamics of the whole S:ACE2 complex with mutations, it is important to consider time scale of microseconds. The authors described the difficulties in terms of computational cost. However, I believe there was a misunderstanding regarding the time scale, yes microsecond but one can consider several parallel runs each of 200 ns. In this way, one can have 5 independent replicas, better statistics and obtain some dynamics of the possible conformations induced by those mutations. As the author mentioned, it will take 6-10 months to runs 1 microsecond in Virginia Tech HPC, well, in practice it may take less than a 1-2 month considering replica study. The suggested parallel study is common in MD.”*

Authors’ response: The authors understand the reviewer’s concern regarding the importance of conformational changes in protein-protein binding. The new suggestion from the reviewer, to conduct replicates of shorter time, differs from the initial request for continuous microsecond trajectories (Reviewer’s #3 comment from initial submission). Capturing large-scale conformational changes in such a large system would likely require continuous simulation for over a microsecond, as originally indicated by the reviewer. Substituting shorter simulations to achieve the same aggregate time would not allow for the observation of the same effects. The suggested approach with replicates would allow for local conformational changes, but most of the time would likely be spent reaching equilibrium rather than true equilibrium sampling.

Previous literature on the ensemble-based simulation of the partial S protein trimer complex has been performed for 7.5 us microseconds, a task that demanded considerable effort and resources (80 P100 GPUs) (Casalino L, et al. AI-driven multiscale simulations illuminate mechanisms of SARS-CoV-2 spike dynamics. *Int J High Perform Comput Appl.* 2021 Sep;35(5):432-451. doi: 10.1177/109434202111006452). Substantial simulation time is necessary to observe S protein RBD conformational dynamics, particularly concerning the impact of mutations at the interface. Simply simulating a single full chain of the S protein would overstate the observed flexibility in the trimer and further would not offer new insights into the relevant binding interface, as suggested by the simulations.

Finally, we consulted with an independent MD simulations expert, Dr. Justin Lemkul, who currently serves as a developer for CHARMM, one of the most popular MD simulation engines. Although, Dr. Lemkul did not contribute directly to this study, he acknowledged that our assertions are justified and fall within the interpretative resolution of the presented data. While the reviewer’s suggestion emphasizes thoroughness, it’s crucial to consider whether such simulations align with the primary focus of this project and whether extensive MD simulations would significantly affect the outcome. We have acknowledged Dr. Lemkul’s expertise in the Acknowledgments section of the manuscript.

Reviewer 3, Comment #3: “the authors now report values of free energy after perform large-scale low-mode conformational sampling which is another form of the simple NMA for large structures. The problem I can see is to have applied to a complex and not a single chain. For the case of the full S:ACE2 complex considering the mutations. The values are very contradictory as now they posses error bars and the absolute values are quite different from each other: E471V-RBD conformations (-151.7 ± 19.2 kcal/mol) and for BA.2-RBD conformations (125.4 ± 4.1 kcal/mol). Here the problem is reflected in the free energy difference, which is ~ -300 kcal/mol between both variants, this is totally off. I believe this is the results of having conformations from NMA which are not consistent with the true conformational landscape of the protein complex.”

Authors’ response: The authors thank the reviewer for identifying this error and sorry that it caused confusion. There was a typographical mistake in the paragraph (Line 243) discussing the large-scale low-mode conformational sampling (please refer to the snapshot of the text below with the typo highlighted). The difference between WT and mutation is only ~ 25 kcal/mol, consistent with the other numerical values. This error has been corrected in the revised version of the manuscript.

~~interactions, the mean ΔG_{bind} for the top five most favorable E⁴⁷¹V_{RBD} conformations (-151.7 ± 19.2 kcal/mol) was more favorable than the mean for BA.2_{RBD} conformations (125.4 ± 4.1 kcal/mol, $p \leq 0.05$). The sampled conformations of E⁴⁷¹V_{RBD} exhibit a~~

Reviewer 4, Comment #1: “TreeLine is not a standard program for phylogenetic (only 41 citations compared to >6500 for IQTree2). I suspect because the trees are not so well resolved, with many polytomies like in Figure 3 and not able to run bootstraps. But this is not a phylogenetics paper and the tree in Figure 3 is not necessarily wrong.”

Authors’ response: The authors thank the reviewer for their positive feedback. We acknowledge the reviewer’s suggestion to try IQTree2 and explore the comparison between tree topologies.

First, the TreeLine function in the DECIPHER package is a relatively new tool for generating phylogenetic trees (Published in April 2022), so it is not expected to have as many publications as more established programs like IQ-Tree (Published in January 2015). We compared the two trees generated using TreeLine (previously in the paper) and IQ-Tree of the same alignment of the sequences used in Figure 3 were remarkably similar to each other including the many polytomies (see output images included at the end of this document). For the IQ-tree the line thickness indicates the bootstrap value. Thicker lines have higher boot strap values. Circled in red are the 3 isolates (highlighted in red boxes below) of 100 samples that were found in different locations between the two trees and each had low bootstrap values. Thus, the trees are of comparable quality. The authors believe that the likely many polytomies seen are due to the sequences lacking enough information to figure out how those they are related. Our wild animal

sample (V0654185) that "moved" is a sequence with ~25% coverage so it lacks sufficient coverage for optimal phylogenetic analysis.

IQTree2 generated for response:

Tree currently in the paper

REVIEWERS' COMMENTS

Reviewer #3 (Remarks to the Author):

I see the authors corrected the confusing values of the free energy difference. I would simply advise the authors to provide the information of results available via zenodo.org. The current link <https://osf.io/82n73/> connected to the website of the authors does not allow for proper verification of the MD data.

REVIEWERS' COMMENTS

Reviewer #3 (Remarks to the Author):

I see the authors corrected the confusing values of the free energy difference. I would simply advise the authors to provide the information of results available via zenodo.org. The current link <https://osf.io/82n73/> connected to the website of the authors does not allow for proper verification of the MD data.

We thank the reviewer for their suggestion. We have uploaded the results to zenodo.org as requested and can be found at the link (<https://zenodo.org/records/11404190>) and DOI [10.5281/zenodo.11404189](https://doi.org/10.5281/zenodo.11404189). These have been added to the data availability statement as well.